# Indoor Airborne VOCs from Water-Based Coatings: Transfer Dynamics and Health Implications [note 1]

**DOI:** 10.3390/jox15060197

**Published:** 2025-12-01

**Authors:** Jana Růžičková, Helena Raclavská, Marek Kucbel, Pavel Kantor, Barbora Švédová, Karolina Slamová

**Affiliations:** 1Centre for Energy and Environmental Technologies, ENET Centre, VSB–Technical University of Ostrava, 17. listopadu 15/2172, 708 00 Ostrava-Poruba, Czech Republic; jana.ruzickova@vsb.cz (J.R.); helena.raclavska@vsb.cz (H.R.); pavel.kantor@vsb.cz (P.K.); barbora.svedova@vsb.cz (B.Š.); 2Institute of Foreign Languages, VSB–Technical University of Ostrava, 17. listopadu 15/2172, 708 00 Ostrava-Poruba, Czech Republic; karolina.slamova@vsb.cz

**Keywords:** volatile organic compounds (VOCs), indoor air quality, polyurethane coatings, acrylate–polyurethane coatings, Sick Building Syndrome (SBS), toxicological risk

## Abstract

Volatile organic compounds (VOCs) emitted from indoor surface coatings can significantly impact indoor air quality and health. This study compared emissions from water-based polyurethane (PUR) and acrylate–polyurethane (ACR–PUR) coatings, identifying 94 VOCs across 16 chemical classes. Time-resolved concentrations were analysed via Principal Component Analysis (PCA)**,** which revealed distinct temporal emission patterns and chemically coherent clusters. Aromatic hydrocarbons, alcohols, esters, and isocyanates dominated the emission profiles, with ACR–PUR releasing markedly higher concentrations of symptom-relevant compounds. Acute exposure was linked to toluene, styrene, phenol, and methyl butyl ketone (MBK), which decreased sharply within 60 days, while compounds such as 1,3-dioxolane, isopropylbenzene, and ethenyl acetate exhibited persistent emissions, suggesting increased chronic risk. Although total VOC levels remained below the German UBA “excellent” threshold (<200 µg/m^3^), neurotoxic and carcinogenic compounds remained detectable. The combination of PCA-based temporal insights with toxicological profiling and emission transfer dynamics offers a refined framework for indoor air risk assessment. These results underscore the need to complement total VOC indices with symptom-oriented, time-resolved screening protocols to better evaluate SBS risk in indoor environments using water-based coatings.

## 1. Introduction

People spend more than 90% of their time indoors, in homes, offices, and commercial spaces [1]. Since the 1970s, research has increasingly focused on the link between indoor environments and health, particularly as the use of synthetic building materials has expanded. Prolonged exposure to indoor air and the growing presence of synthetic products have made indoor air quality a key determinant of public health. Indoor air is a complex mixture of chemical and biological contaminants, including gases, volatile and semi-volatile organic compounds (VOCs and SVOCs), particulate matter, and microorganisms such as mites, moulds, bacteria, and viruses. These pollutants can significantly reduce environmental quality and adversely affect human health. Their sources include everyday activities such as heating, cooling, humidification, cooking, cleaning, and smoking.

Among indoor pollutants, VOCs in both gaseous and particulate forms are particularly significant. They are frequently associated with Sick Building Syndrome (SBS) [2] and Sick House Syndrome [3], conditions characterised by a spectrum of health complaints linked to inadequate ventilation and elevated indoor temperatures. The release profile of VOCs depends strongly on their chemical composition, volatility, and stability under environmental factors such as temperature, UV radiation, and relative humidity [4]. For instance, floor coatings mainly emit alcohols and esters [5], while wood lacquers release a broader spectrum of VOCs, including aromatic hydrocarbons (toluene, xylenes, ethylbenzene, styrene), aldehydes (formaldehyde, hexanal), alcohols (n-butanol, ethylene glycol), terpenes, and higher molecular-weight hydrocarbons [6].

Modern water-based coatings, particularly acrylic-polyurethane, polyurethane, and polyester-polyurethane systems, are widely used for finishing wood, furniture, toys, and decorative surfaces. Their appeal lies in their mechanical resistance, adhesion, UV durability, and more favourable ecological properties compared to solvent-based coatings. These systems rely on polymer dispersions in water and are assumed to reduce VOC emissions. Nevertheless, water-based coatings can still emit harmful VOCs and SVOCs. Their release dynamics and stability under environmental conditions such as temperature, humidity, and UV exposure remain insufficiently studied, especially regarding their contribution to indoor air quality. Wang et al. [7], for example, demonstrated that VOC concentrations and odour intensity increased with temperature, while higher humidity and ventilation reduced them. The ratio of air exchange to surface load was also shown to significantly influence VOC emissions, underscoring the importance of proper storage and application conditions.

Within the European Union, VOC content in coatings is regulated under Directive 2004/42/EC [8], which sets maximum concentrations for products in their ready-to-use form. However, this regulation only addresses product formulation and does not account for emission behaviour over time. The duration and magnitude of VOC release from surface treatments remain unregulated. Post-application emission behaviour is typically evaluated only within voluntary certification schemes and ecolabels such as the EU Ecolabel or Blue Angel, which require VOC measurements, generally after 28 days, and set limits for total emissions, release rates, or the presence of specific hazardous compounds.

This regulatory gap highlights the need for systematic investigation of how water-based polyurethane (PUR) and acrylic–polyurethane (ACR–PUR) coatings behave once applied indoors, not only in terms of total VOC levels but also in the toxicological relevance and persistence of individual compounds. Addressing this gap, the present study integrates chemical composition data, emission profiles, and time-resolved indoor air measurements to quantify the transfer efficiency of toxicologically classified substances and identify those with the highest potential for acute symptoms and chronic exposure.

A novel framework for VOC risk assessment is introduced by calculating the percentage transfer of individual compounds into indoor air at two post-application intervals (14–21 days and 60 days). Unlike conventional approaches based solely on emission factors or total VOC (TVOC) thresholds, this method enables the identification of compounds with high acute relevance (such as phenol, methyl butyl ketone, and vinyl acetate) as well as long-term persistence (such as 1,3-dioxolane and isopropylbenzene). The findings support the design of time-resolved screening panels that account for both toxicological classification and emission dynamics, contributing to more targeted indoor air quality management strategies. This study aimed to identify and characterise volatile and semi-volatile organic compounds (VOCs and SVOCs) released from water-based coatings at different post-application stages (14–21 days and 60 days) and evaluate their toxicological relevance in relation to indoor air quality.

Unlike previous studies that primarily focused on total VOC levels or emission factors from water-based coatings, this work introduces a compound-specific assessment framework that integrates emission dynamics with toxicological relevance. The approach compares the relative release of individual VOCs and SVOCs into indoor air at different post-application stages (14–21 days and 60 days), enabling the identification of both acutely relevant and long-term persistent compounds. By linking time-resolved emission behaviour with toxicological classification, this study provides a more realistic understanding of the exposure potential of coating-derived pollutants. This framework extends conventional TVOC-based evaluations and supports the development of targeted strategies for indoor air quality management.

## 2. Materials and Methods

### 2.1. Materials

Volatile and semi-volatile organic compound (VOC, SVOC) emissions were investigated from two commercially available waterborne coatings manufactured in the Czech Republic, representing polyurethane (PUR) and acrylic–polyurethane (ACR–PUR) dispersion varnishes. Both coatings are intended for interior wooden substrates such as flooring, parquet, furniture, and other indoor surfaces.

The PUR coating is a one-component, water-based varnish formulated for professional use on indoor wooden surfaces. The declared composition includes isothiazolinone-based preservatives and benzotriazole derivatives (poly(oxy-1,2-ethanediyl), α-[3-[3-(2H-benzotriazol-2-yl)-5-(1,1-dimethylethyl)-4-hydroxyphenyl]-1-oxopropyl]-ω-hydroxy-). According to the manufacturer, the product has a VOC content of 0.075 kg/kg, a maximum volatile content of ≤80 g/L, and a density of 1.05 g/cm^3^. The ACR–PUR coating is a waterborne acrylic–polyurethane varnish designed for long-term durability in indoor applications and marketed as safe and environmentally friendly. It complies with EN 71-9 [9], confirming suitability for surfaces in contact with children’s toys. Hazardous constituents listed include 5-chloro-2-methylisothiazol-3(2H)-one and 2-methylisothiazol-3(2H)-one (3:1). The declared VOC content is 0.065 kg/kg, with a maximum volatile content of ≤70 g/L, and a density of 1.05 g/cm^3^. Although the manufacturer declared isothiazolinone-based preservatives and benzotriazole derivatives as hazardous constituents, these compounds were not detected in the VOC emission profiles due to their low volatility. Instead, the emission spectrum was dominated by solvents, co-solvents, plasticisers, and degradation products. Comparative properties of the coatings are summarised in Table 1.

Waterborne PUR varnishes are typically designed for high mechanical resistance and durability. Their VOC emission profile is relatively stable, releasing fewer compounds under standard conditions, although latent reactive residues such as unreacted isocyanates may be mobilised under accelerated environmental stress. In contrast, ACR–PUR coatings often exhibit higher initial volatility, attributable to acrylate-derived esters and aldehydes that volatilise more readily. This difference explains the elevated levels of symptomatically relevant VOCs reported for acrylate-containing systems [10].

### 2.2. Sampling Procedure

Semi-volatile organic compounds (SVOCs) represent an important class of indoor pollutants with physicochemical properties intermediate between VOCs and particulate-bound compounds. Despite their lower vapour pressure, SVOCs are continuously emitted from various indoor sources—including building materials, plasticisers, coatings, electronic equipment, and consumer products—via slow volatilisation, abrasion, or thermal release. Once emitted, these compounds partition between the gas phase, airborne particles, and settled dust, resulting in prolonged residence times and multiple exposure pathways.

The sorbent tubes used in this study (Tenax TA, Carbograph 1TD, and Carboxen 1003) (Markes International, Ltd., Bridgend, UK) enable simultaneous sampling of both VOCs and SVOCs, which allowed for the detection of compounds such as phthalate esters, polycyclic aromatic hydrocarbons (PAHs), and cyclic siloxanes. The presence of these compounds in the gaseous phase has been confirmed in multiple studies [11,12,13,14]. Although their volatility is limited, their persistence and continuous low-level emissions make SVOCs relevant for indoor exposure assessment. In the context of Sick Building Syndrome, such compounds may contribute to chronic irritation, fatigue, or other non-specific symptoms through long-term exposure and accumulation in indoor environments.

Emissions of VOCs and SVOCs were collected both directly above the opened varnish cans and from indoor air after application. For sampling above the liquid surface, sorption tubes (Markes International, Ltd., Bridgend, UK) connected to an Acti-VOC pump (Markes International, Ltd., Bridgend, UK) were used, with a collection time of 5 min. As no standard specifies the sampling distance for headspace collection directly from product containers, the tube inlet was placed approximately 1–2 cm above the surface of the liquid. This arrangement allowed for the capture of primary emissions released immediately after opening. Each varnish sample was analysed in duplicate. To distinguish emissions originating from coating materials from those potentially released by the wooden substrate, a two-step experimental procedure was applied. In the first step, emissions were measured directly from the liquid coating. A sorbent tube connected to an ACTI-VOC pump was positioned approximately 1–2 cm above the open container of the coating material, allowing for the characterisation of its intrinsic VOC/SVOC emission profile. This measurement represents a material emission characterisation, not an exposure scenario, and was performed under controlled conditions following the standard headspace or source emission sampling approach [15,16,17]. In the second step, air sampling was performed in the test room where wooden panels had been coated with the same materials and allowed to dry under controlled conditions. The difference between the emission spectra obtained from the coated wood (Step 2) and the coating-only baseline (Step 1) was interpreted as the contribution of the wooden substrate. The derived emission levels were compared with literature data for untreated wood and showed good agreement. This approach corresponds to the analytical principle of a standard addition method, which enables the identification and quantification of compound sources within complex indoor environments.

To isolate the relative contributions of coatings and the wooden substrate to VOC and SVOC emissions, a standard addition approach was applied. Emissions originating solely from the coatings were first determined by direct sampling of vapours from sealed containers containing only the coating materials, using sorbent tubes for TD-GC/MS analysis. Subsequently, chamber experiments were conducted in which wooden panels coated with the same materials were placed in the test room, and VOC/SVOC concentrations were measured under identical conditions. The difference between the emission profiles obtained from the coatings alone and those from the coated wood surfaces represents the contribution of the wooden substrate. This relative contribution was further verified by comparison with published emission data for untreated wood under similar temperature and humidity conditions.

The manufacturer declares that the tested coatings reach “zero emissions” after 14 days of curing. The objective of this study was to verify whether emissions can still be detected beyond this declared period and to characterise their persistence over time. Therefore, air sampling was performed between day 14 and day 21 after coating application, and again on day 60. These intervals represent post-curing stages during which emissions are expected to be minimal according to product specifications.

The initial high-emission phase occurring within the first days after application was not included, as it is well documented in previous studies and was outside the scope of this research, which focused on long-term and residual emission potential of the cured coatings.

Indoor air sampling was conducted in a test room (Figure 1) with a total volume of 75 m^3^, in accordance with ISO 16017-2 [18]. Indoor temperature and relative humidity were continuously monitored (dynamically) with a Govee WiFi H5179 sensor (Shenzhen Intellirocks Tech Co., Ltd., Shenzhen, Guangdong, China), which has an accuracy of ±0.3 °C for temperature and ±3% RH for relative humidity. The coated surface area, which included wall cladding, beams, and flooring, was 80 m^2^. Indoor samples were first collected 14 days after coating application, corresponding to the manufacturer’s recommendation for achieving the declared performance properties. Sampling then continued daily during the following eight days, that is, from day 14 to day 21. Three additional control samples were taken after 60 days to evaluate long-term emissions. Sampling was carried out at a room temperature of 20 ± 2 °C, while outdoor daily temperatures ranged between 5 and 7 °C. Air samples were collected with the Acti-VOC pump onto sorption tubes containing a multi-bed packing of Tenax TA, Carbograph 1TD, and Carboxen 1003 (Markes International, Ltd., Bridgend, UK), ensuring a broad spectrum of VOCs and SVOC retention. Each indoor sampling event lasted 1 h.

Three Acti-VOC pumps (Figure 1) were placed inside the chamber to enable parallel sampling. Sorbent tubes were positioned in the breathing zone (approximately 1.2–1.5 m above the floor) and at least 1 m away from walls, windows, doors, or large objects, as recommended by ISO 16000-5 [19], which defines the sampling strategy for volatile organic compounds in indoor environments. This placement was also consistent with the approach used by Marzocca et al. [20]. Ventilation of the test room relied exclusively on natural air exchange through the window opening. Nonmechanical ventilation was used. The room where the coating was applied was ventilated manually five times per day to simulate realistic intermittent air exchange conditions typical of naturally ventilated interiors. Such a regime prevents excessive VOC accumulation while avoiding complete removal of emissions, thereby allowing the observation of representative indoor air concentrations and their temporal evolution. Each ventilation cycle consisted of opening the windows for approximately 5–10 min, resulting in a short-term increase in air exchange rate followed by a gradual re-equilibration of indoor concentrations. The chosen frequency corresponded to an average air exchange rate of approximately 0.5–1.0 h^−1^, consistent with values reported for residential and office spaces under moderate user activity [16,21]. This controlled ventilation schedule ensured a reproducible balance between emission retention and dilution, enabling meaningful interpretation of the temporal dynamics of VOC release from the tested coatings.

Based on the measured concentrations of chemical compounds during daily sampling, the arithmetic mean and standard deviation were calculated for each compound. Compounds with a relative standard deviation (RSD) exceeding 30% were excluded from further analysis. The 30% RSD threshold was selected based on commonly accepted criteria for analytical repeatability in VOC sampling studies (e.g., ISO 16000 series, United States Environmental Protection Agency (U.S. EPA): Compendium Method TO-17: Determination of Volatile Organic Compounds in Ambient Air Using Active Sampling onto Sorbent Tubes). The resulting values were then used to evaluate daily changes in concentration.

### 2.3. Analytical Methods

Collected sorbent tubes were analysed by thermal desorption–gas chromatography/mass spectrometry (TD-GC/MS; Gerstel, Mülheim an der Ruhr, Germany). An internal standard (1,3,5-tri-tert-butylbenzene) was added to each tube before desorption. The desorption program consisted of heating from 50 °C (1 min) to 300 °C (5 min) at the rate of 60 °C/min. The analytes were cryofocused in a cooled injection system (CIS) at 10 °C, then rapidly heated at 10 °C/s to 250 °C and transferred to a non-polar HP5 ms column (60 m × 250 µm × 0.25 µm). The GC temperature program started at 40 °C (10 min), increased at 15 °C/min to 300 °C, and was held for 10 min.

Identification and quantification of VOCs and SVOCs were performed using authentic reference standards. Indoor temperature and relative humidity were continuously monitored with a Govee WiFi H5179 sensor (Shenzhen Intellirocks Tech Co., Ltd., Shenzhen, Guangdong, China), which has an accuracy of ±0.3 °C for temperature and ±3% RH for relative humidity. These data provided additional context for evaluating emission dynamics under variable microclimatic conditions.

### 2.4. Statistical Evaluation

The dataset comprised time-resolved concentrations of volatile organic compounds (VOCs) emitted from polyurethane (PUR) and acrylate–polyurethane (ACR–PUR) lacquers. Each compound was quantified at ten time points: eight measurements between day 14 and day 21 after application, and two additional measurements on day 60. Basic descriptive statistics (arithmetic mean and standard deviation) were computed in OriginPro (OriginLab Corporation, Northampton, MA, USA).

Before multivariate analysis, the data were arranged into a matrix of dimensions (samples × variables), where “samples” represented either measurement days or individual VOCs, depending on the analytical perspective. Missing values were replaced with the mean value of the respective variable. This means that for any missing data point in a given variable, the mean of the non-missing values for that variable was used as a proxy. This approach was used because the percentage of missing data was small (<5%), and it helps avoid excluding incomplete data from the analysis while maintaining the overall dataset structure. To eliminate scale effects arising from differences in absolute concentration levels among compounds, all data were standardised using a z-score transformation. Principal Component Analysis (PCA) was performed on the correlation matrix of the standardised dataset. Two complementary perspectives were applied. In the compound-centred approach, rows represented individual VOCs and columns represented measurement days, yielding factor loadings that identified groups of compounds contributing most strongly to each principal component. In the day-centred approach, rows corresponded to measurement days and columns to compounds, resulting in PCA scores that enabled the evaluation of temporal trends in emission profiles. PCA computations and visualisations were implemented in Python 3.14 software environment with pandas 2.3, NumPy 2.1, scikit-learn 1.7, and Matplotlib 3.9 libraries (Python Software Foundation, Wilmington, DE, USA). Where relevant, the number of retained components was guided by the scree plot (elbow), the Kaiser criterion (eigenvalues > 1), and cumulative explained variance, with five components ultimately selected for interpretation.

#### Statistical Robustness and Component Retention Criteria

To verify the robustness of the PCA solution, several complementary validation approaches were applied. First, a cross-validation procedure based on the leave-one-out method confirmed that the loadings and explained variance of the retained components (PC1–PC5) remained stable within ±3% across iterations. The Kaiser criterion (eigenvalue > 1) and inspection of the scree plot both supported the retention of five components, as higher-order PCs (PC6 and above) contributed only marginally (<1.5%) and lacked consistent chemical meaning. In addition, the loadings of the dominant VOCs within each PC were highly consistent across bootstrapped subsamples (r > 0.95), indicating strong internal coherence.

Each of the five retained components also exhibited a distinct and physically interpretable chemical profile. PC1 corresponded to solvent diffusion, PC2 to glycol-ether evaporation, PC3 to slow phthalate migration, PC4 to aldehydic oxidation processes, and PC5 to phenolic stabiliser release. This correspondence between statistical structure and chemical processes confirms the robustness and interpretability of the PCA solution, ensuring that the extracted components represent genuine temporal and chemical trends rather than artefacts of data variability.

### 2.5. Potential Risk Assessment of Chemical Substances in Relation to Sick Building Syndrome (SBS)

To evaluate the health relevance of emissions from the tested varnishes in relation to Sick Building Syndrome (SBS), all identified substances were classified according to internationally recognised schemes and authorities, including the European Chemicals Agency (ECHA), Regulation (EC) No 1272/2008 of the European Parliament and of the Council of 16 December 2008 on classification, labelling and packaging of substances and mixtures, amending and repealing Directives 67/548/EEC and 1999/45/EC, and amending Regulation (EC) No 1907/2006 (Text with EEA relevance) (CLP), the Globally Harmonized System (GHS), the International Agency for Research on Cancer (IARC), the U.S. Environmental Protection Agency (U.S. EPA), and Organisation for Economic Co-operation and Development (OECD) guidance documents. The assessment focused on toxicological endpoints most relevant to SBS symptoms: carcinogenicity, reproductive toxicity, mutagenicity, neurotoxicity, acute toxicity, skin and eye irritation, and specific target organ toxicity (STOT). Harmonised classifications were taken from the CLP Annex VI and the ECHA database. Where harmonised data were not available, Safety Data Sheets (SDS) provided by manufacturers or suppliers were used for verification.

Acute toxicity, irritation, and STOT were prioritised as they correspond to immediate and non-specific symptoms often reported indoors, including eye and skin discomfort, mucosal irritation, headaches, fatigue, and respiratory complaints. The evaluation combined toxicological potency with the measured indoor air concentrations, enabling the classification of each compound into three SBS relevance categories: high, medium, or low. This framework allowed for the identification of substances with the greatest potential to trigger or exacerbate SBS (Table 2).

Table 2 does not originate from a single published source; instead, it was compiled by the authors as a synthesis of internationally recognised toxicological classification systems and regulatory frameworks. These include the CLP Regulation (EC No 1272/2008), GHS, IARC, U.S. EPA, ECHA guidance documents, and OECD Test Guidelines. The table was created to align toxicological endpoints with symptoms commonly associated with SBS, based on classifications and criteria defined in these sources. We have clarified this in the manuscript.

The relevance of individual compounds to SBS was evaluated using a multi-criteria expert approach. Each compound was assessed according to: (i) toxicological potency (irritation, sensitisation, chronic toxicity/STOT; data from ECHA/CLP);(ii) SBS association and exposure evidence (reported links to SBS symptoms, measured indoor air concentrations, and frequency of detection across studies; sources such as Wolkoff et al. [17], Nazaroff and Weschler [22]; Norbäck et al. [23]; Ait Bamai et al. [24]); (iii) physicochemical parameters influencing volatility and phase distribution (vapour pressure, boiling point, log Kₒ_w_, water solubility; obtained from PubChem).

The three evidence streams were qualitatively weighted, giving greater importance to compounds combining higher toxicological potency and frequent detection in indoor air. The initial ranking was supported by AI-assisted literature screening, with expert review ensuring the final category assignments (strong, moderate, weak) were internally consistent and evidence-based.

## 3. Results and Discussion

### 3.1. Characterisation of Organic Compounds in Indoor Air and Water-Based Varnishes

In the analysed samples of indoor air and water-based varnishes (polyurethane and acrylic-polyurethane), sixteen groups of organic compounds were identified (Appendix A). These included additives, degradation products, precursors, resin monomers (polyester, polyurethane, acrylic), contaminants, and secondary compounds formed through interactions between varnishes and wood. Additives represented a substantial fraction, fulfilling technological functions such as photoinitiators, antioxidants, UV stabilisers, plasticisers, drying agents, adhesion promoters, gloss enhancers, and antimicrobial agents. They were detected in indoor air either in their original form or as degradation and reaction products. Their presence and concentrations depended on the type of varnish, resin composition, and interactions with the underlying wood.

In the air inside the test chamber, three main groups of organic compounds dominated: aromatic hydrocarbons (37.40 ± 12.90%) > alcohols (23.64 ± 8.74%) > carboxylic acids, esters, and acetates (10.5 ± 2.90%). Water-based varnishes were a significant source of alcohols and esters, which are widely used as solvents and resin components [25]. The wooden substrate also contributed to the overall burden of aromatic hydrocarbons (9–11%), with elevated emissions of toluene and 1,3-xylene from untreated wood reported by Wang et al. [25].

Alcohols represented the second most abundant group. Their concentrations in emissions from PUR and ACR–PUR varnishes were comparable, reflecting their role as solvents in both systems. Carboxylic acids, esters, and acetates accounted for about 10% of the total. According to Alapieti et al. [26], coatings significantly alter the emission profile compared to bare wood, with esters and acetates becoming dominant during early drying stages. Their release is influenced by substrate moisture. These compounds typically occur at 11–20% in coating formulations as part of polyester and acrylic resins modified with polyurethane and serve as plasticisers and lubricants.

Cyclic alkenes, mainly terpenes and isoprenoids, were released naturally from raw wood. They are constituents of resins and essential oils, especially in coniferous species. However, when coatings such as polyurethane varnishes were applied, processes occurred that altered the volatility of these compounds [27]. The varnish could dissolve wood constituents, enhance their release, or promote chemical interactions between terpenes (e.g., α-pinene, limonene) and varnish components, which modify the final emission profile. Depending on the type of varnish, some VOCs were trapped in the coating layer, while others were more intensively emitted into indoor air.

Aldehydes and ketones originated from surface treatments, construction materials, and furniture, acting as secondary pollutants. Major sources included MDF, particle boards, laminates, and adhesives. Liu et al. [27] reported mean concentrations of aldehydes and ketones in 30 monitored indoor environments at 0.432 mg/m^3^, with the highest values for flooring (0.648 mg/m^3^), furniture (0.590 mg/m^3^), and coatings (0.341 mg/m^3^).

The total VOC concentration in the test chamber was 389.33 ± 30.96 µg/m^3^ during days 14–21 after coating application. For the evaluation of varnish impact on indoor air quality, only compounds of anthropogenic origin and with proven health risks were considered, while terpenes and isoprenoids were excluded. In total, 96 organic compounds with potentially toxic properties were identified, several of which have been associated with Sick Building Syndrome symptoms (Appendix A).

#### 3.1.1. Reproductive Toxicants

Many substances with established reproductive toxicity were detected in indoor air following the application of PUR and ACR–PUR varnishes. These included phthalates such as bis(2-ethylhexyl)phthalate (DEHP), dibutyl phthalate (DBP), and diethyl phthalate (DEP), glycol ethers such as 2-ethoxyethanol (EGEE), solvents such as methyl butyl ketone (MBK/2-hexanone) and N,N-dimethylformamide (DMF), as well as styrene, toluene, and 1-ethyl-2-methylbenzene.

Among these, DEHP was detected at 7.46 ± 5.01 µg/m^3^ in chamber air during days 14–21, while concentrations in the polyurethane varnish reached 62.66 ± 11.33 µg/m^3^. After 60 days, indoor levels dropped below the detection limit. DEHP is classified as a presumed human reproductive toxicant (Repr. 1B) and as a probable human carcinogen by the U.S. EPA [28]. Its toxicological profile includes disruption of thyroid hormone balance, reproductive organs, the liver, and the nervous system [29]. Although DEHP itself is not considered a direct SBS-inducing compound, its degradation product 2-ethyl-1-hexanol has been repeatedly associated with SBS symptoms in buildings with PVC flooring [30].

DBP was present at 6.37 ± 2.09 µg/m^3^ in PUR and 1.94 ± 0.10 µg/m^3^ in ACR–PUR varnish. In indoor air, its concentration averaged 0.16 ± 0.14 µg/m^3^ during days 14–21 but decreased markedly to 0.03 ± 0.002 µg/m^3^ after 60 days. This rapid decline is consistent with its higher volatility and lower molecular weight compared to DEHP, which facilitates faster evaporation and weaker binding to the polymer matrix. Despite its faster loss, DBP is a recognised endocrine disruptor and has been shown to exacerbate allergen-induced lung function decline and alter airway immunology [31].

DEP, a less potent phthalate, was detected at 6.37 ± 2.09 µg/m^3^ in PUR and 1.94 ± 0.1 µg/m^3^ in ACR–PUR varnish. Chamber concentrations averaged 0.44 ± 0.15 µg/m^3^ during days 14–21 and declined to 3.1 ± 0.1 ng/m^3^ after 60 days [32]. Despite lower toxicity, its presence illustrates that even regulated or phased-out plasticisers can contribute to indoor VOC burdens.

Solvents with reproductive toxicity were also present. MBK (2-hexanone) was measured at 9.59 ± 1.42 µg/m^3^ in PUR and 4.97 ± 0.14 µg/m^3^ in ACR–PUR varnish. Chamber concentrations were stable at 0.61 ± 0.14 µg/m^3^ during days 14–21 and 0.60 ± 0.15 µg/m^3^ after 60 days, demonstrating persistence due to intermediate volatility and reversible sorption–desorption processes. DMF, used as a solvent to improve diisocyanate solubility in PUR systems, was present at 31.16 ± 11.21 µg/m^3^ in ACR–PUR varnish and detected in chamber air at 0.22 ± 0.07 µg/m^3^ during days 14–21, but declined below the detection limit after 60 days. Its rapid loss reflects its high vapour pressure and susceptibility to hydrolysis and photodegradation.

Styrene and toluene, both common VOCs in varnishes and wooden composites, were also detected. Styrene reached 26.49 ± 1.20 µg/m^3^ in PUR and 188.65 ± 87.25 µg/m^3^ in ACR–PUR varnishes, while chamber air concentrations were 3.39 ± 1.22 µg/m^3^ during days 14–21 and 1.13 ± 0.017 µg/m^3^ after 60 days. Toluene exhibited extreme variability between formulations, with 2.45 ± 3.21 µg/m^3^ in PUR but as high as 816.65 ± 215.1 µg/m^3^ in ACR–PUR varnish. In chamber air, its concentrations averaged 30.19 ± 11.45 µg/m^3^ in days 14–21 but fell below detection by day 60, consistent with its high volatility and photochemical reactivity.

Finally, 1-ethyl-2-methylbenzene, a by-product of varnish manufacture and a respiratory irritant, was detected at 38.28 ± 4.67 µg/m^3^ (PUR) and 25.27 ± 3.92 µg/m^3^ (ACR–PUR) in varnishes, with indoor concentrations of 0.33 ± 0.16 µg/m^3^ during days 14–21 and non-detectable after 60 days. Similarly, EGEE (2-ethoxyethanol), a glycol ether solvent, was measured at 1.21 ± 0.14 µg/m^3^ (PUR) and 6.45 ± 0.97 µg/m^3^ (ACR–PUR), while indoor levels declined from 0.090 ± 0.026 µg/m^3^ (days 14–21) to 0.044 ± 0.020 µg/m^3^ (day 60).

Overall, reproductive toxicants displayed different emission behaviours: some (e.g., toluene, DMF) declined rapidly, while others (e.g., MBK, phthalates) persisted for weeks. This mixture of transient and long-lasting compounds underlines the complexity of indoor exposure scenarios and their relevance to both acute and chronic SBS symptoms.

#### 3.1.2. Mutagens in Indoor Air from Polyurethane and Acryl-Polyurethane Lacquers

Polyurethane and acryl–polyurethane lacquers can release mutagenic compounds into indoor environments, particularly benzene, glyoxal, and phenol. Benzene, a well-known mutagen and human carcinogen, originates both from natural components of wood, such as lignin and extractives, and from varnish composition. Studies have reported emissions of up to 12.82 µg/m^3^ from untreated solid wood [25], while concentrations in our test materials were substantially higher: 37.56 ± 11.89 µg/m^3^ in PUR varnish and 32.73 ± 6.83 µg/m^3^ in ACR–PUR varnish. In chamber air, benzene levels reached 28.42 ± 14.98 µg/m^3^ during days 14–21, exceeding the average background values in European households of 8.42 ± 14.98 µg/m^3^ [33]. By day 60, however, concentrations had dropped below detection, consistent with observations by Kumar et al. [34] that VOCs from coatings rapidly decline due to volatilisation, photochemical degradation, and dilution.

Phenol is another mutagenic compound frequently found indoors, originating from coatings, adhesives, carpets, plastics, and wooden furniture. Reported background levels range from 0.2 to 21.5 µg/m^3^ [35]. In our measurements, phenol emissions from PUR varnish reached 46.04 ± 29.23 µg/m^3^, while ACR–PUR varnish produced 37.64 ± 11.90 µg/m^3^. Indoor chamber concentrations were much lower but still notable at 4.53 ± 1.97 µg/m^3^ (days 14–21), followed by a sharp decline to 0.06 ± 0.004 µg/m^3^ after 60 days. This reduction is in accordance with systematic reviews indicating that VOC emissions from coatings and building materials can diminish by 90–99% within two months of application [34].

Glyoxal, used as a cross-linking agent in polymers and also applied as a biocide, was identified at 10.63 ± 0.97 µg/m^3^ in ACR–PUR varnish. Chamber concentrations averaged 0.35 ± 0.09 µg/m^3^ during days 14–21, comparable to residential indoor levels of 0.42 µg/m^3^ reported by Duncan et al. [36]. In more oxidative indoor environments, however, glyoxal has been detected at ~2 µg/m^3^ [37]. Its absence by day 60 reflects its high reactivity and instability: as a small α-dicarbonyl compound, glyoxal undergoes rapid heterogeneous reactions with nucleophilic functional groups, sorption to surfaces, and transformation into secondary products such as imines, acetals, or oligomers. This behaviour explains its relatively short persistence in indoor air without continued emission sources.

#### 3.1.3. Carcinogens and Potential Carcinogens in Indoor Air from Polyurethane and Acryl–Polyurethane Lacquers

Several carcinogenic or potentially carcinogenic compounds were detected in association with PUR and ACR–PUR varnishes, including 1,4-dioxane, acetamide, vinyl acetate, isopropylbenzene, p-aminotoluene, tetrachloroethylene, benzophenone, and residual polymeric methylene-diphenyl diisocyanate (RMDI). As discussed in Section 3.1.2. benzene is included as one of the most relevant agents.

1,4-Dioxane, a possible human carcinogen, may occur as a residual solvent or degradation product of polyether components in varnishes. It was detected in PUR varnish at 1.69 ± 0.26 µg/m^3^ and at identical levels in chamber air during days 14–21. By day 60, concentrations had declined to 0.053 ± 0.002 µg/m^3^, approaching background values reported for households in Japan (0.02 µg/m^3^) [38]. Importantly, the EU-US risk screening level for long-term cancer risk is 0.2 µg/m^3^ [39], meaning that concentrations during the early phase exceeded health-based thresholds, though they dropped to safe levels after two months.

Acetamide, a possible by-product of synthesis or an additive, was present at ~4–5 µg/m^3^ in both varnish types. Chamber concentrations were modest (0.073 ± 0.052 µg/m^3^ during days 14–21) but increased slightly to 0.123 ± 0.003 µg/m^3^ after 60 days. Such delayed release is atypical but consistent with diffusion-limited behaviour in highly cross-linked polymer matrices and with potential secondary indoor sources, such as MDF panels and linoleum [40].

Vinyl acetate, used in hybrid polymer binders, was measured at 1.80 ± 0.21 µg/m^3^ in PUR and 13.17 ± 8.27 µg/m^3^ in ACR–PUR varnishes. Indoor chamber concentrations rose from 0.621 ± 0.48 µg/m^3^ (days 14–21) to 1.534 ± 1.09 µg/m^3^ after 60 days. This trend mirrors findings by Huang et al. [37] and reflects the slow release of polymer-bound volatiles under realistic indoor conditions.

Isopropylbenzene (cumene), a precursor for methyl methacrylate, was present at 7.67 ± 0.19 µg/m^3^ in PUR and 1.63 ± 0.70 µg/m^3^ in ACR–PUR varnishes. In chamber air, concentrations stabilised around 0.44 ± 0.18 µg/m^3^ (days 14–21) and 0.40 ± 0.01 µg/m^3^ (day 60), reflecting its relatively low volatility, high boiling point, and chemical stability.

p-Aminotoluene, an aromatic amine and known carcinogenic intermediate in pigment and polyurethane production, was found at 0.61 ± 0.05 µg/m^3^ in PUR varnish and 0.062 ± 0.026 µg/m^3^ in chamber air during days 14–21. By day 60, concentrations fell below detection, consistent with literature values for indoor environments [41].

Tetrachloroethylene, widely used as a solvent in coatings, was present at ~25 µg/m^3^ in both PUR and ACR–PUR varnishes. Chamber air levels were much lower, averaging 0.062 ± 0.046 µg/m^3^ during days 14–21 and 0.040 ± 0.010 µg/m^3^ after 60 days. Despite rapid initial volatilisation, its persistence suggests slow desorption from indoor surfaces such as textiles and wall materials, aligning with results reported by Fromme et al. [33].

Benzophenone, a photoinitiator and possible human carcinogen, was detected at ~20–25 µg/m^3^ in both varnish types. In chamber air, concentrations were 0.043 ± 0.013 µg/m^3^ during days 14–21 but dropped below detection by day 60. This pattern is consistent with previous studies showing its partitioning onto dust particles and resuspension in indoor air [42].

Finally, polymeric methylene-diphenyl diisocyanate (RMDI), classified by ECHA as a suspected human carcinogen (Category 2), was detected as multiple stereoisomers in indoor air. Between days 14–21, total concentrations reached 0.108 ± 0.028 µg/m^3^, dominated by the cis,trans isomer at 0.042 ± 0.007 µg/m^3^. Emissions were substantially higher in ACR–PUR varnishes (403.88 ± 42.76 µg/m^3^) than in PUR varnishes (64.05 ± 13.76 µg/m^3^), reflecting differences in prepolymerisation degree and residual monomer content [43]. By day 60, however, all isomers were below detection, suggesting progressive degradation, sorption, and diffusion processes typical of reactive isocyanates.

In summary, these findings confirm that both mutagens and carcinogens are relevant components of varnish emissions (Figure 2). Although many compounds decline rapidly within two months, several exceed health-based guidance levels during the early post-application period, highlighting the importance of ventilation and material selection in indoor environments.

### 3.2. Toxicological Profile of PUR and ACR–PUR Coatings

The comparative analysis of PUR and ACR–PUR coatings demonstrates apparent differences in composition and toxicological relevance of the emitted compounds. These variations are critical for assessing health risks associated with indoor exposure (Table 3).

ACR–PUR coatings exhibited a higher neurotoxic potential, primarily due to elevated levels of 1,3-dioxolane (42.72 ± 31.05 µg/m^3^), compared to PUR (73.36 ± 10.75 µg/m^3^). This compound is associated with central nervous system effects and sensory irritation, indicating a strong neurobehavioural relevance of ACR–PUR emissions.

Reproductive toxicants also showed contrasting profiles. ACR–PUR contained markedly higher concentrations of styrene (188.65 ± 87.25 µg/m^3^) and toluene (816.65 ± 215.10 µg/m^3^), both linked to endocrine disruption and developmental toxicity. PUR, on the other hand, was characterised by higher levels of phthalates, particularly bis(2-ethylhexyl) phthalate (62.66 ± 11.33 µg/m^3^), known to impair fertility and foetal development. These divergent pathways indicate that both coatings present reproductive risks, though via different mechanisms.

Mutagenic compounds were detected in both formulations. Benzene was dominant in PUR (37.56 ± 11.89 µg/m^3^), while glyoxal was more prominent in ACR–PUR (10.63 ± 0.97 µg/m^3^). Phenol was common to both, further contributing to the mutagenic burden under prolonged exposure conditions.

Carcinogenic and potentially carcinogenic substances displayed the strongest contrast. ACR–PUR contained significantly higher levels of reactive methylene diphenyl diisocyanate (RMDI) isomers, such as cis,trans- (213.17 µg/m^3^) and cis,cis- (153.10 µg/m^3^), which are associated with tumorigenic outcomes. Tetrachloroethylene (24.09 ± 7.65 µg/m^3^) and benzophenone (24.56 ± 1.28 µg/m^3^) were also more abundant in ACR–PUR, reinforcing its elevated carcinogenic potential.

These results indicate that ACR–PUR coatings pose a substantially greater toxicological burden across all evaluated categories. While PUR coatings are comparatively less hazardous, the presence of persistent compounds such as phthalates and benzene requires attention in long-term exposure scenarios.

### 3.3. Toxicological Assessment of Indoor Air Following Coating Application

While coating-specific profiles highlight intrinsic differences in toxicological potential, the actual exposure risk is determined by the temporal evolution of compounds in indoor air (Table 4).

The neurotoxicant 1,3-dioxolane exhibited remarkable stability, with concentrations of 2.46 ± 0.90 µg/m^3^ between day 14 and day 21 and 2.47 ± 0.16 µg/m^3^ by day 60. This persistence underlines its significance for chronic exposure scenarios, especially in poorly ventilated environments.

Reproductive toxicants showed the highest absolute load immediately after application, with toluene at 33.94 ± 11.45 µg/m^3^ and styrene at 3.39 ± 1.22 µg/m^3^. By day 60, toluene fell below the detection limit, whereas styrene remained detectable at 1.13 ± 0.02 µg/m^3^. Phthalates such as bis(2-ethylhexyl) phthalate (7.46 ± 5.01 µg/m^3^) were confined to the early phase.

Mutagenic substances such as benzene (28.42 ± 14.98 µg/m^3^) and phenol (4.53 ± 1.97 µg/m^3^) almost completely disappeared by day 60, with total concentrations falling to 0.06 ± 0.004 µg/m^3^. This rapid reduction indicates their volatile nature and acute rather than chronic relevance.

Carcinogenic and potentially carcinogenic compounds decreased only modestly, from 1.43 ± 0.21 µg/m^3^ at day 14 to 21 to 1.27 ± 0.11 µg/m^3^ at day 60. Some, such as ethenyl acetate, even increased slightly, from 0.62 ± 0.48 µg/m^3^ to 1.53 ± 1.09 µg/m^3^, probably due to desorption or secondary formation processes. Acetamide (0.073 ± 0.052 → 0.123 ± 0.003 µg/m^3^) and isopropylbenzene (0.44 ± 0.18 → 0.40 ± 0.01 µg/m^3^) persisted at comparable levels, while RMDI isomers were confined to the early phase.

### 3.4. Integrated Evaluation of Emission Behaviour and Exposure Relevance

Neurotoxicants such as 1,3-dioxolane maintained a stable transfer efficiency of 3.37% between day 14 and day 21 (2.46 ± 0.90 µg/m^3^ from 73.36 ± 10.75 µg/m^3^) and 3.39% at day 60 (2.47 ± 0.16 µg/m^3^ from 72.72 ± 11.40 µg/m^3^). This persistence confirms their potential for long-term neurobehavioural effects.

Reproductive toxicants showed the highest absolute emission load (1460 ± 121 µg/m^3^), but their transfer efficiency was relatively low: 3.25% between day 14 and day 21, falling to 0.30% by day 60. MBK remained comparatively stable (0.61 ± 0.14 µg/m^3^ versus 0.60 ± 0.15 µg/m^3^), whereas phthalates, toluene, and styrene declined sharply. These findings highlight the dominance of acute risks in the early phase and a limited set of persistent contributors to long-term exposure.

Mutagenic compounds displayed high early transfer rates (37.16%), particularly phenol (63.51%), but by day 60, their concentrations nearly disappeared (0.06 ± 0.004 µg/m^3^). This confirms their acute relevance and limited persistence in indoor air.

Carcinogens initially had low transfer rates, about 0.24% of total emissions, but several persisted over time. Ethenyl acetate increased slightly from 0.62 ± 0.48 µg/m^3^ to 1.53 ± 1.09 µg/m^3^, while acetamide and isopropylbenzene remained nearly constant. RMDI isomers and most aromatic carcinogens were present only in the early phase.

Quantitatively, aromatic hydrocarbons and alcohols accounted for more than 60% of the total VOC reduction between days 14–21 and day 60, confirming their key role in the early emission phase and their high volatility shortly after application. Esters and phthalates contributed significantly to the overall decline, although their decrease was more gradual, likely due to sorption on surfaces or within the coating matrix. Groups with marginal contributions, such as amines, amides, and ethers, showed either low initial concentrations or limited volatility, thereby reducing their overall influence on the emission balance. From the quantified compounds, 83.2% of the total VOC reduction was captured, while the remaining 16.8% represented residual concentrations that were either not degraded or had fallen below the detection limit. These substances are often characterised by low reactivity or long-term persistence and are critical for assessing chronic exposure and designing targeted filtration measures, particularly in spaces with insufficient ventilation.

Acute risks are primarily associated with highly volatile substances released shortly after coating application, while chronic exposure is shaped by low-volatility compounds with persistent emission profiles. This dual pattern underscores the importance of time-resolved assessment for effective indoor air quality management.

### 3.5. Sick Building Syndrome

The concept of Sick Building Syndrome (SBS) remains a matter of debate, as it represents a descriptive set of symptoms—such as headaches, eye and skin irritation, and fatigue—that can arise from multiple interacting factors, including inadequate ventilation, chemical and biological contaminants, and psychosocial conditions [44]. The present study does not establish chemical causality but explores how certain volatile and semi-volatile organic compounds (VOC/SVOC) released from coating materials may contribute to the complex environmental exposures associated with SBS. This approach provides insight into potential chemical contributors without excluding other influencing factors such as ventilation efficiency or occupant sensitivity.

Sick Building Syndrome (SBS) refers to a set of health problems reported by occupants of specific buildings without the identification of a distinct illness or single causative factor. As described by Norback et al. [45], SBS has a multifactorial origin, reflecting a complex interaction of chemical, physical, and psychosocial conditions. Rather than being attributable to a single pollutant, SBS arises from a combined exposure to volatile organic compounds (VOCs), inadequate indoor climate parameters (such as temperature and humidity), poor lighting, noise, occupational stress, and individual sensitivity. The heterogeneity of human response is significant, since compounds that may be tolerated by one individual can cause irritation or systemic effects in another.

#### 3.5.1. Comparison of Coatings

The chemical compositions of emissions from polyurethane (PUR) and acryl–polyurethane (ACR–PUR) coatings show pronounced differences that can be directly linked to their SBS potential (Table 5). Polyurethane lacquer (PUR) is characterised by a relatively high content of alcohols (994.3 ± 135.6 µg/m^3^), consistent with a traditional solvent system of lower reactivity. Its emission profile also reveals lower levels of aromatic hydrocarbons (112.8 ± 34.2 µg/m^3^), which are typically associated with neurotoxicity and sensory irritation, and only limited concentrations of isocyanates (64.0 ± 18.3 µg/m^3^). Such a profile suggests either a less reactive formulation or an efficient curing process with minimal residual monomers. From the perspective of SBS, this corresponds to a coating with reduced potential to trigger acute sensory symptoms, although alcohols and diols (254.8 ± 42.7 µg/m^3^) still contributed substantially to overall VOC load and odour perception. Such residual emissions are consistent with the secondary diffusion phase described by Alapieti et al. [26] for polyurethane dispersions, but the persistence observed here was longer, indicating slower volatilisation under real indoor conditions. In contrast, ACR–PUR lacquer exhibited a more complex and aggressive emission spectrum. Elevated concentrations of aromatic hydrocarbons (946.2 ± 175.4 µg/m^3^), esters (280.8 ± 51.2 µg/m^3^), diols (412.7 ± 63.5 µg/m^3^), ethers (198.3 ± 40.1 µg/m^3^), and isocyanates (403.9 ± 72.6 µg/m^3^) were observed, reflecting a multi-component system modified with acrylates. These modifications, while improving the mechanical and optical properties of the coating, also enhanced the volatility and reactivity of the formulation, which significantly increased its emission potential and toxicological burden. As a result, ACR–PUR coatings can be expected to pose a greater risk for SBS symptom onset, especially in enclosed spaces with limited ventilation. Similar emission patterns have been reported for hybrid acrylate–polyurethane formulations [27], but the emission magnitudes observed here exceed those typical for “low-emission” products, confirming that manufacturer declarations are not always representative of actual indoor conditions.

Although the manufacturer classifies the tested ACR–PUR coatings as “low-emission”, measurable levels of toxicity-relevant VOCs were still detected after the declared emission-free period. Similar inconsistencies between marketing claims and actual emission behaviour have been documented in previous studies [46,47], where even water-based or “eco-labelled” coatings were shown to release a complex mixture of VOCs under realistic indoor conditions. This highlights that low-emission labelling does not necessarily guarantee negligible emissions, particularly when coating formulations include reactive or slowly evaporating components. This observation is further supported by Uhde and Salthammer [48], who emphasised that reaction products derived from indoor materials, regardless of their marketed emission profile, may negatively impact indoor air quality due to their low odour thresholds and health-related properties.

Figure 3 illustrates these contrasts quantitatively. While alcohols dominated in PUR coatings (994.3 ± 135.6 µg/m^3^), the ACR–PUR formulation was characterised by much higher levels of aromatic hydrocarbons (946.2 ± 175.4 µg/m^3^) and isocyanates (403.9 ± 72.6 µg/m^3^). This marked shift in the dominant compound groups directly correlates with a higher symptomatic potential, since aromatic and isocyanate compounds are among the most frequent triggers of SBS-related irritation and respiratory complaints [23]. A detailed breakdown of all identified substances and their classification into symptom relevance categories is presented in Appendix A. As shown there, PUR emissions are dominated by compounds of moderate relevance. In contrast, ACR–PUR contains a greater proportion of substances falling into strong and moderate-to-strong categories, highlighting its elevated symptomatic potential despite being marketed as a low-emission coating.

Figure 3 confirms that PUR coatings, despite their lower overall aggressiveness, still contain compounds capable of provoking irritation during prolonged exposure. In contrast, ACR–PUR coatings present a substantially higher SBS risk due to the elevated presence of strongly symptomatic chemical groups, as evidenced by the dominance of aromatic hydrocarbons and isocyanates.

#### 3.5.2. Indoor Air Quality After Coating Application

A total of 94 compounds were identified and classified into 16 chemical groups based on their structural characteristics and known associations with Sick Building Syndrome (SBS) symptoms. The distribution of these compounds by chemical class and their relative SBS relevance is shown in Figure 4. The test chamber contained no additional materials or furnishings, which allowed for a clear attribution of the measured concentrations to the coatings applied.

Most detected compounds were alcohols, aromatic hydrocarbons, and carboxylic esters, reflecting their role as primary components of waterborne coatings. These groups are strongly or moderately associated with SBS-related symptoms, including mucosal irritation, neurotoxicity, and sensory discomfort. Aldehydes, diols, ethers, and amines were also present at relevant levels, indicating secondary emissions from degradation or polymerisation processes during film curing. This observation aligns with degradation dynamics reported in previous studies [49,50], but the early appearance of aldehydes in our data indicates that oxidative processes begin sooner than expected after coating application. In contrast, groups such as isocyanates, isocyanides, nitroalkanes, alkanes, and heterocyclic hydrocarbons were represented only by individual compounds. Despite their low numbers, their toxicological significance and reactivity justify their inclusion in acute SBS screening, particularly in poorly ventilated environments.

Figure 4 illustrates the total number of chemical compounds with SBS relevance in indoor air 14–21 days after coating application, and their persistence after 60 days. At the earlier interval, compounds with strong or moderate-to-high SBS associations were dominant, confirming their role in acute symptom onset. Overall, 94 compounds were identified, of which 17 showed strong or moderate-to-strong association with SBS symptoms. Since all detected substances originated from a single emission source, it was possible to evaluate their emission behaviour and symptomatic relevance with higher accuracy and without interference from other materials. By day 60, most of these substances had declined substantially, but compounds with moderate association remained at detectable levels, representing a long-term exposure concern.

The overall quantity of chemical compounds with SBS relevance in indoor air 14–21 days after coating application, and their persistence after 60 days is illustrated in Figure 5. At the earlier interval, compounds with strong or moderate-to-high SBS associations were dominant, confirming their role in acute symptom onset. By day 60, most of these substances had declined substantially, but compounds with moderate association remained at detectable levels, representing a long-term exposure concern.

Quantitative analysis confirmed this temporal pattern. Compounds with strong SBS association decreased from 37.03 ± 2.94 µg/m^3^ at 14–21 days to 2.94 ± 0.41 µg/m^3^ at 60 days, corresponding to a 92% reduction. Substances with moderate-to-high SBS association showed an even steeper decline of 99%, from 49.51 ± 0.33 µg/m^3^ to 0.33 ± 0.07 µg/m^3^. In contrast, compounds classified as moderate exhibited only a 67% reduction (48.16 ± 1.60 µg/m^3^ to 16.04 ± 0.98 µg/m^3^), confirming their persistence and chronic relevance. Compounds with low SBS association decreased by 89% (45.91 ± 0.50 µg/m^3^ to 4.95 ± 0.22 µg/m^3^). These values illustrate that while acute irritants dissipate relatively quickly, chronic contributors such as aldehydes, ketones, and low-volatility esters continue to affect air quality. Similar persistence of aldehydic compounds was reported in field studies by Nazaroff and Weschler [22] and Huang et al. [11].

The greatest relative reductions were observed for phthalate esters (99.1%), phenols (98.8%), diols (98.4%), aromatic hydrocarbons (97.7%), and carboxylic esters (89.8%), consistent with their high volatility and acute emission profiles. In contrast, aldehydes and ketones showed the lowest reduction (32%), reflecting their stability and persistent contribution to indoor air quality. Aldehydes, which are typically formed as secondary oxidation products during polymer ageing, were among the most persistent compounds detected.

This dual emission pattern—a rapid early release of highly volatile irritants followed by the slower persistence of less volatile but toxicologically relevant compounds—confirms that acute SBS symptoms are driven by short-term emissions, whereas chronic effects are sustained by persistent compounds such as aldehydes, ketones, and low-volatility esters. These findings underscore the need for time-resolved SBS risk classification that accounts not only for chemical type but also for emission dynamics [17,23].

In addition, groups such as isocyanates, isocyanides, nitroalkanes, and alkanes were monitored. These compounds showed very low initial concentrations and declined below the detection limit of the analytical method after 60 days. Although their absolute values were minimal, their high reactivity, toxicity, and irritation potential justify their inclusion in the acute SBS screening panel, particularly in poorly ventilated environments or under cumulative VOC exposure conditions [12,13].

Aromatic hydrocarbons and alcohols accounted for more than 60% of the total VOC reduction between days 14–21 and day 60, confirming their key role in the early emission phase and their high volatility shortly after application. Figure 6 illustrates the relative decrease in concentrations of individual chemical groups between days 20 and 60 and their contribution to the overall reduction in VOC burden. Esters and phthalates contributed significantly to the decline, although their decrease was more gradual, likely due to sorption on surfaces or within the coating matrix. Groups with marginal contributions, such as amines, amides, and ethers, showed either low initial concentrations or limited volatility, thereby reducing their overall influence on the emission balance. From the quantified compounds, 83.2% of the total VOC reduction was captured, while the remaining 16.8% represented residual concentrations that were either not degraded or had fallen below the detection limit. These substances—often characterised by low reactivity or long-term persistence—are critical for the assessment of chronic exposure and for the design of targeted filtration measures, particularly in spaces with insufficient ventilation.

Compared with international indoor air standards, the total VOC (TVOC) values measured 14–21 days after coating application reached 180.61 ± 60.6 µg/m^3^. According to the German UBA guideline values [51], this corresponds to excellent air quality (<200 µg/m^3^). The result is also in accordance with the WHO [52] recommendations, which consider TVOC levels below 300 µg/m^3^ as indicative of good indoor air quality. However, despite compliance with these guideline values, the presence of specific neurotoxic, reprotoxic, and irritant compounds at measurable concentrations suggests that the indoor environment may still trigger SBS-related symptoms in sensitive individuals. This finding supports the view expressed by Wolkoff [53], who emphasised that TVOC alone is not a reliable predictor of health outcomes, since SBS symptoms are more closely linked to the toxicological profiles of individual compounds than to their total concentration.

Direct comparison of the measured indoor concentrations with occupational exposure limits (OELs) has limited interpretive value, as these limits (e.g., OSHA—Occupational Safety and Health Administration (U.S. Department of Labor); NIOSH—National Institute for Occupational Safety and Health (U.S.); ACGIH—American Conference of Governmental Industrial Hygienists) are defined for 8 h work exposures in industrial settings. In residential, office, or public buildings, people are typically exposed for much longer periods—often 16–24 h per day—and include more sensitive populations such as children, the elderly, or patients. No legally binding exposure limits exist for such environments; instead, the World Health Organisation (WHO) recommends that the total concentration of volatile organic compounds (TVOC) in indoor air should remain below 300 µg/m^3^ to ensure acceptable air quality. For instance, the occupational exposure limit for toluene varies between 50–200 mg/m^3^ depending on the regulatory body [54,55,56], while indoor concentrations reported in the literature are typically in the range of 10–200 µg/m^3^ [17,57,58]. Although these concentrations are far below workplace thresholds, the much longer exposure duration in homes and offices, combined with the presence of multiple VOCs, may still contribute to Sick Building Syndrome (SBS) through additive or synergistic effects [17,59].

This highlights the need for exposure assessment frameworks that consider chronic, low-level exposure to complex VOC mixtures in non-industrial indoor environments.

#### 3.5.3. Principal Component Analysis (PCA) of VOC Profiles and Their Relevance to SBS

Temporal differentiation of VOC profiles

Principal component analysis (PCA) was performed for the individual measurement days, starting on day 14 after coating and ending on day 21, with additional measurements taken on day 60 after application. PCA is based on the assessment of variance, with each principal component defined by the proportion of variance it explains in the dataset. Here, variance represents the variability of VOC concentrations across the individual measurement days. Five principal components (PC1–PC5) were retained, together explaining 98.5% of the total variance in the dataset, while higher-order components contributed only marginally. The first principal component (PC1) explained 56.0% of the variance, while the second (PC2) accounted for 21.4%. Together, PC1 and PC2 represented 77.4% of the dataset variability. The third, fourth, and fifth components explained 10.1%, 7.9%, and 3.1% of the variance, respectively.

The interpretation of the five principal components (PC1–PC5) revealed distinct chemical processes governing VOC emissions from the tested coatings. PC1 (solvent fraction) showed high scores on days 14–16 and 18, reflecting residual concentrations of toluene, xylene, styrene, and heptane. Since measurements began two weeks after application, these values indicate persistent solvent diffusion from deeper layers of the coating rather than freshly applied vapours. Very low PC1 scores in the 60-day samples confirmed that most of the solvent fraction had dissipated. Similar post-curing diffusion patterns have been described for waterborne coatings by Alapieti et al. [26], but the values observed here were higher, indicating a slower curing process in the ACR–PUR formulation.

PC2 (glycol ethers and co-solvents) became more prominent on days 18 and 20, with higher contributions of 2-butoxyethanol and 2-butoxyethoxyethanol. These compounds, characterised by lower volatility and slower release dynamics, emerged after the main solvent fraction had declined, while low scores on days 14–15 confirmed that glycol ethers were not dominant in the early emission profile. This temporal shift corresponds to the delayed release phase observed for glycol ether-based formulations [17], yet our data demonstrate that this contribution remains significant even beyond 20 days, highlighting prolonged exposure potential in closed environments.

PC3 (phthalate plasticisers) showed slightly elevated values on day 21 and in the 60-day samples, consistent with the very slow migration of phthalates from the polymer matrix. The results confirm the long-term persistence of plasticisers such as bis(2-ethylhexyl) phthalate and dibutyl phthalate, which appear in the gas phase only after extended ageing. This pattern complements field observations reported by Huang et al. [11] and Olkowska et al. [12], but the earlier onset observed in our study suggests faster diffusion under test-chamber conditions. PC4 (aldehydic degradation products) increased on days 19, 21, and 60, indicating secondary formation of aldehydes such as hexanal, nonanal, benzaldehyde, and octanal. These compounds are known indicators of oxidative ageing in coatings [22]. Our data confirm that oxidation becomes relevant already within the first three weeks, with aldehydes subsequently dominating the long-term emission phase. Finally, PC5 (phenolic stabilisers and antioxidants) showed increased scores on days 20, 21, and 60, linked to the release of compounds such as phenol, 2,4-di-tert-butylphenol, and BHT. Their presence at later stages indicates that stabilisers incorporated into the coating formulation undergo gradual desorption and degradation. This phenomenon has been associated with polymer ageing [49]; however, our data show that such degradation may begin earlier and reach higher relative concentrations than typically assumed for “low-emission” coatings.

High PC1 scores on days 15–17 and 19 reflect residual concentrations of toluene, xylene, styrene, and heptane (Figure 7). As the first measurement corresponds to day 14 after application, these values indicate persistent emissions of solvents diffusing from deeper coating layers rather than freshly applied vapours. Very low PC1 scores in the 60-day samples confirm that most of the solvent fraction had dissipated. Positive PC2 scores on days 18 and 20 indicate a stronger contribution of 2-butoxyethanol and 2-butoxyethoxyethanol, which volatilise more slowly and therefore become prominent after the solvent fraction has diminished. Low PC2 scores on days 14–15 confirm that glycol ethers did not dominate the early emission profile. The biplot thus highlights the contrasting dynamics of solvents (PC1) and glycol ethers (PC2).

The heatmap (Figure 7) extends this interpretation by incorporating all five principal components. It reveals later-stage processes, including phthalate migration (PC3, days 21 and 60), aldehyde formation (PC4, days 19, 21, and 60), and phenolic stabiliser release (PC5, days 20, 21, and 60). Together, the two visualisations provide a comprehensive fingerprint of the temporal evolution of VOC emissions.

PCA loadings, dominant VOC contributors, and their relevance to SBS

In our dataset, the first two principal components (PC1 and PC2) explained 56.0% and 21.4% of the total variance, respectively, accounting together for 77.4%. The third to fifth components added 10.1%, 7.9%, and 3.1%, so that the first five altogether captured 98.5% of the total variability. Higher-order components (PC6 and beyond) contributed only marginally and typically represented noise or subtle effects without stable interpretability. Based on both their explanatory power and chemical meaning, five components were retained for interpretation. PC1 represented the solvent fraction of the coating, defined mainly by toluene, o-xylene, styrene, 1,2,4-trimethylbenzene, and heptane. PC2 reflected glycol ethers and their esters, such as 2-butoxyethanol, 2-butoxyethoxyethanol, 1-methoxypropanol, and 1-(1-methoxypropan-2-yloxy)propan-2-yl acetate, typical co-solvents and rheological modifiers. PC3 was linked to phthalate plasticisers, including bis(2-ethylhexyl) phthalate, diethyl phthalate, and dibutyl phthalate, indicative of slow migration from the polymer matrix. PC4 captured aldehydic degradation products (hexanal, nonanal, benzaldehyde, and octanal), reflecting oxidative ageing of the coating film. Finally, PC5 was associated with phenolic antioxidants and stabilisers (phenol, 2,4-di-tert-butylphenol, butylated hydroxytoluene), representing additives released during later stages of degradation (Table 6). Together, these components describe the progression from primary solvent emissions to the slower release of co-solvents, plasticisers, and degradation products, providing a comprehensive picture of VOC dynamics in the studied coatings. The contribution of individual compounds to the PCA, expressed as a heat map, is shown in Figure 8.

The five principal components (PC1–PC5) identified in the dataset were further interpreted in relation to their toxicological and sensory properties, providing a framework to assess their relevance for Sick Building Syndrome (SBS). PC1, dominated by solvent-related emissions, and PC2, defined by glycol ethers, were strongly linked to acute SBS symptoms such as irritation, headaches, and neurotoxic effects. These findings are consistent with the symptom patterns described by Wolkoff et al. [17] and Norbäck et al. [23], but the relative weighting of solvents and glycol ethers derived from PCA demonstrates that both contribute comparably to short-term sensory discomfort. PC4, characterised by aldehydic oxidation products, was also associated with acute complaints, particularly sensory irritation and the perception of stale air. In contrast, PC3 (phthalates) and PC5 (phenolic compounds) reflected long-term or secondary contributions, including odour nuisance, allergic responses, and endocrine-disrupting potential, as indicated by the persistence of these groups in the late emission phase [11,24,60].

The network analysis (Figure 9) illustrates the links between the principal components identified by PCA and specific symptoms of Sick Building Syndrome (SBS). Solvent-related compounds (PC1), glycol ethers (PC2), and aldehydes (PC4) are predominantly associated with acute effects such as irritation, headache, and odour perception. In contrast, phthalates (PC3) and phenolic antioxidants (PC5) contribute more to chronic discomfort, odour nuisance, and allergic responses. This visualisation highlights how groups of VOCs defined by PCA translate into distinct health-relevant symptom clusters.

This integrative interpretation of PCA results provides a clear framework linking emission processes with their toxicological and sensory relevance. The data demonstrate that while acute SBS symptoms are primarily associated with solvent and glycol ether emissions, chronic discomfort and potential systemic effects arise from the slower release of plasticisers and degradation products. Such temporal differentiation underscores the importance of time-resolved analysis when assessing indoor air quality, as also highlighted by Wolkoff [61] and the WHO [52].

## 4. Conclusions

This study shows that acrylate-modified polyurethane (ACR–PUR) coatings emit substantially higher levels of toxicologically relevant VOCs than conventional PUR coatings, thereby increasing the risk of Sick Building Syndrome (SBS). Although the coatings were applied and ventilated according to the manufacturer’s instructions, detectable VOC and SVOC emissions persisted beyond the declared emission-free period. This persistence highlights the need to verify low-emission product claims under realistic indoor conditions and to consider residual emissions in indoor air quality assessments. Highly volatile compounds such as toluene, styrene, and benzene dominated the early emission phase and were linked to acute symptoms, while less volatile compounds, including 1,3-dioxolane, methyl butyl ketone (MBK), ethenyl acetate, and phthalates, persisted longer and posed a greater risk of chronic exposure.

Principal Component Analysis (PCA) revealed a temporal progression from solvents shortly after application to co-solvents, plasticisers, and degradation products at later stages, providing mechanistic insight into emission dynamics. These findings underline the need to evaluate VOCs not only by total concentrations but also by their temporal behaviour and toxicological profiles. While PUR coatings may be relatively safer, they still present risks under prolonged exposure or insufficient ventilation. Future studies should extend this approach to real-world indoor environments with multiple emission sources and interacting materials to better capture cumulative health impacts.

Understanding how emissions from building materials contribute to perceived indoor air quality and occupant well-being will be crucial for developing safer materials and evidence-based ventilation guidelines.

## Figures and Tables

**Figure 1 jox-15-00197-f001:**
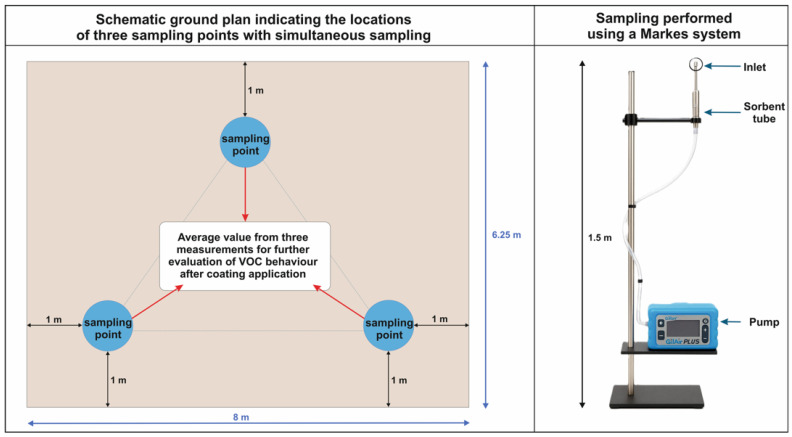
The sampling procedure and site.

**Figure 2 jox-15-00197-f002:**
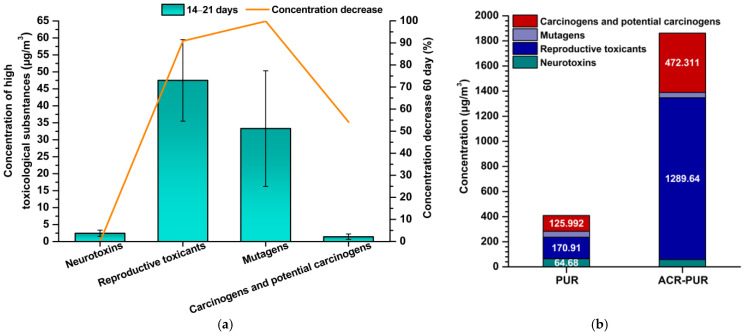
Concentrations of toxicologically relevant compounds 14–21 days after coating application and their reduction after 60 days (**a**), detected concentrations of identical compounds in emissions from coating materials (**b**).

**Figure 3 jox-15-00197-f003:**
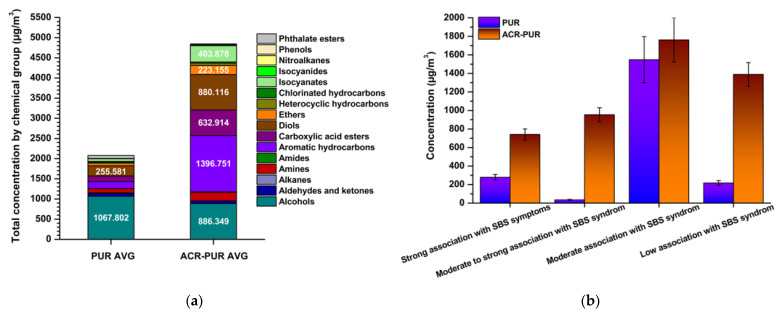
Total concentrations of chemical compounds by chemical group (**a**); distribution of chemical compounds in relation to SBS potential (**b**).

**Figure 4 jox-15-00197-f004:**
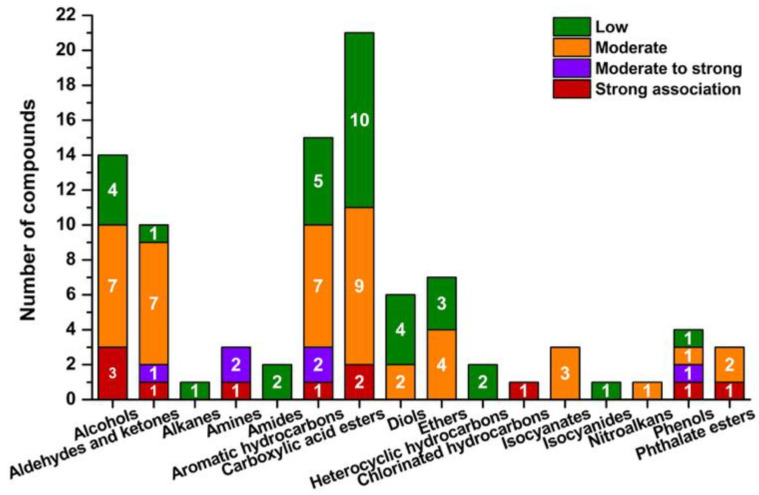
Number of chemical compounds associated with SBS by organic group.

**Figure 5 jox-15-00197-f005:**
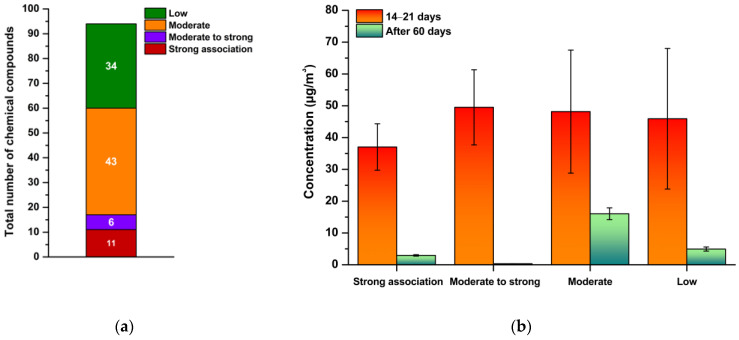
The total number of chemical compounds associated with SBS in indoor air, (**a**) 14–21 days after coating application; (**b**) VOC profile grouped by SBS risk potential.

**Figure 6 jox-15-00197-f006:**
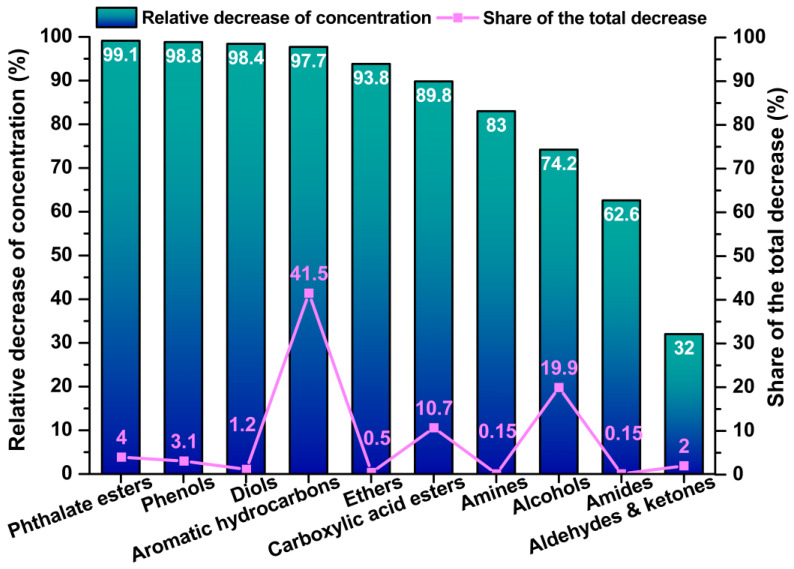
Relative VOC reduction and group contribution.

**Figure 7 jox-15-00197-f007:**
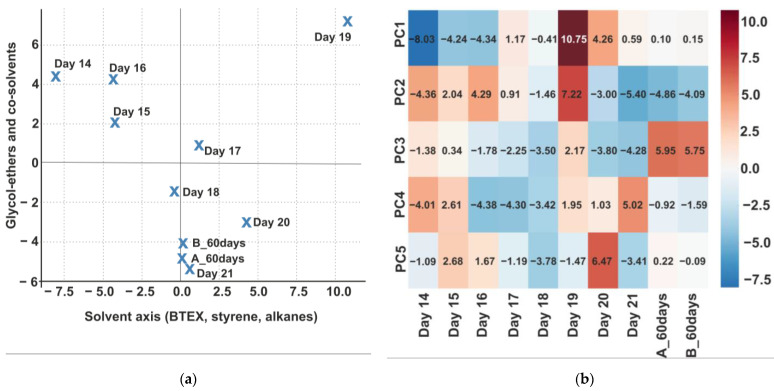
Temporal differentiation of VOC profiles based on PC1 and PC2 (**a**), heatmap of PCA scores (PC1–PC5) across measurement days (**b**).

**Figure 8 jox-15-00197-f008:**
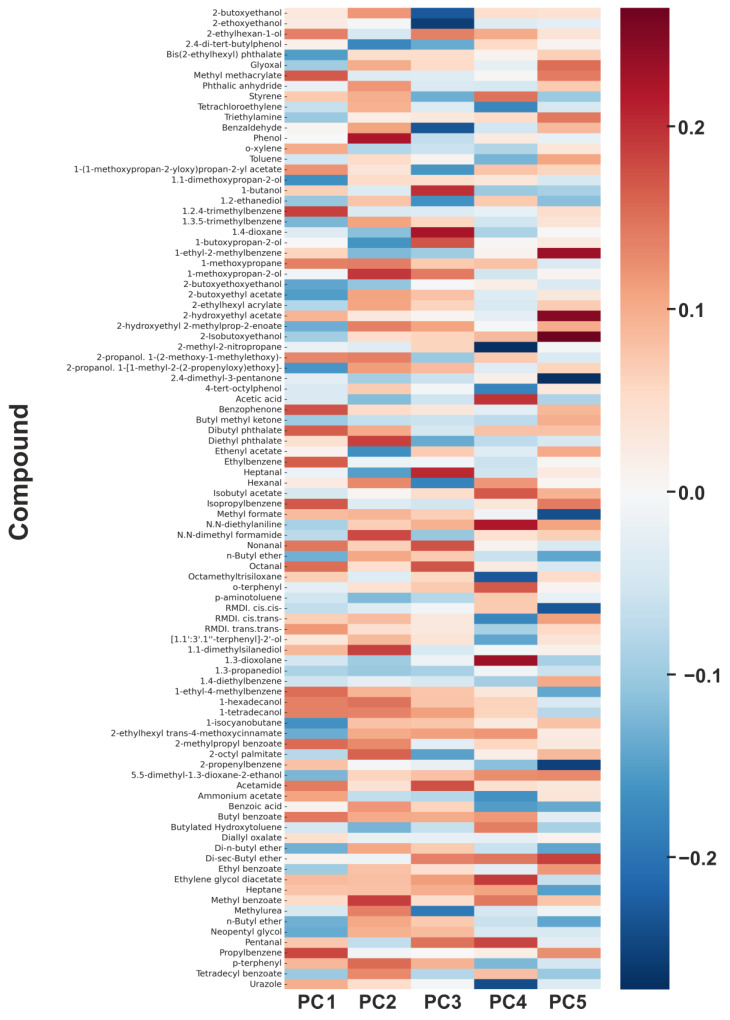
Heat map showing all compounds and their relationship to the principal components (PCA).

**Figure 9 jox-15-00197-f009:**
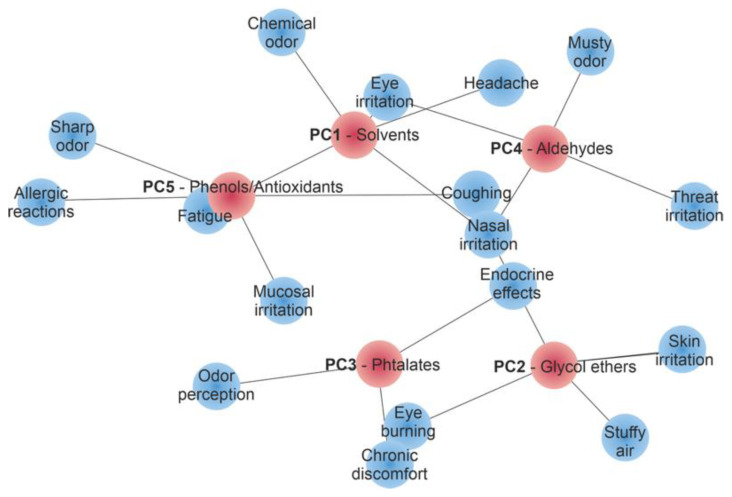
Network visualisation of principal components (PC1–PC5, red nodes) and their associated Sick Building Syndrome (SBS) symptoms (blue nodes). Edges indicate literature-based or toxicological links between compound groups, defining each principal component and reported SBS manifestations.

**Table 1 jox-15-00197-t001:** Comparative properties of the tested water-based coatings.

Property	Water-Based PUR Coating	Water-Based ACR–PUR Coating
Binder	Polyurethane dispersion	Blend of acrylates and polyurethane
VOC profile	Lower; may contain reactive residues (e.g., isocyanates)	Higher initial volatility due to acrylate esters and aldehydes
Emission dynamics	More stable, slower VOC release	Faster and more intense VOC release
UV resistance	Lower	Higher (due to acrylate component)
Application properties	Longer drying time; requires precise application	Faster drying, better adhesion

**Table 2 jox-15-00197-t002:** A toxicological classification framework applied for the risk assessment of SBS-related substances.

Toxicological Category	Classification Source/Authority	Criteria and Scope	Examples of Classification Codes/Endpoints	Relevance to SBS Symptoms
Carcinogenicity	CLP, IARC-U.S. EPA	Substances classified as Carcinogenicity 1A, 1B, 2 based on human or animal data	Carc. 1A (proven), Carc. 2 (suspected)	Chronic risk from prolonged exposure
Reproductive toxicity	CLP/GHS, Annex I Section 3.7	Effects on fertility, development, and lactation	Repr. 1A, 1B, 2	Fertility, developmental, and endocrine effects
Mutagenicity (germ cell)	CLP/GHS, Annex I Section 3.5	Induction of heritable mutations in germ cells	Muta. 1A, 1B, 2	Genetic stability, chronic SBS effects
Neurotoxicity	ECHA guidance, OECD TG 426, TG 443	Developmental and adult neurotoxicity, DNT cohorts	Based on DNT data, not codified in CLP	Headache, dizziness, cognitive impairment
Acute toxicity	CLP Annex VI, ECHA, SDS	Systemic effects after single exposure via oral, dermal, and inhalation	AT1–AT4, NC; H301, H302, H312, H330, H331, H332	Nausea, dizziness, and respiratory distress
Skin/eye irritation	CLP/GHS, SDS	Local effects, reversible or irreversible	Skin 1/1A/1B, Skin 2, Eye 1, Eye 2/2A/2B	Burning eyes, mucosal irritation, and skin discomfort
Specific Target Organ Toxicity (STOT)	CLP/GHS Annex I Section 3.8	Organ effects after single (SE) or repeated (RE) exposure	STOT SE 1–3, STOT RE 1–2	Burning eyes, mucosal irritation, and skin discomfort

**Table 3 jox-15-00197-t003:** Key toxicological groups and dominant VOCs by coating type.

Group	Key Compounds	Coating with a Higher Concentration
Neurotoxins	1,3-Dioxolane	Acrylate–polyurethane
Reproductive toxicants	Toluene, styrene, phthalates	Acrylate–polyurethane (toluene, styrene), Polyurethane (phthalates)
Mutagens	Benzene, glyoxal	Polyurethane (benzene), Acrylate–polyurethane (glyoxal)
Carcinogens	Residual polymeric methylene-diphenyl diisocyanate (RMDI) isomers, tetrachloroethylene	Acrylate–polyurethane

**Table 4 jox-15-00197-t004:** Temporal behaviour of toxicologically relevant compounds in indoor air post-coating application.

Group	Key Compounds	Temporal Behaviour
Neurotoxins	1,3-Dioxolane	Stable, persistent
Reproductive toxicants	Toluene, styrene, phthalates	Sharp decline; styrene—partly persistent
Mutagens	Benzene, phenol	Near-complete reduction
Carcinogens	Ethenyl acetate, acetamide, RMDI isomers	Modest decline; some persistent

**Table 5 jox-15-00197-t005:** Major chemical groups associated with SBS risks in PUR and ACR–PUR coatings.

Chemical Group	SBS-Related Risk	Dominant in
Amines	Sensitisation, odour	ACR–PUR
Aromatic hydrocarbons	Neurotoxicity, irritation	ACR–PUR
Isocyanates	Respiratory sensitisation	ACR–PUR
Esters	Mucosal irritation	ACR–PUR
Alcohols	High volatility, odour	PUR
Phenols	Mucosal irritation	ACR–PUR

**Table 6 jox-15-00197-t006:** Top 10 compounds with the highest loadings for each of the five principal components (PC1–PC5).

Component	Top 10 Contributing Compounds (Loading)
PC1 Solvent fraction	Octanal (−0.396), Hexanal (0.393), 2-butoxyethoxyethanol (0.389), Styrene (−0.379), 2-butoxyethanol (0.372), o-xylene (−0.364), Nonanal (0.351), 1,2,4-trimethylbenzene (−0.342), 1-methoxypropan-2-ol (0.335), Benzaldehyde (0.328)
PC2 Glycol ethers and their esters	1-methoxypropan-2-ol (0.402), Heptane (0.401), o-xylene (0.397), Styrene (0.383), 2-butoxyethanol (−0.372), Hexanal (−0.361), Phenol (0.348), 2,4-di-tert-butylphenol (0.342), Nonanal (−0.334), 1,2,4-trimethylbenzene (0.329)
PC3 Co-solvents and rheological modifiers	Toluene (−0.526), Bis(2-ethylhexyl) phthalate (0.498), Diethyl phthalate (0.482), Dibutyl phthalate (0.467), Nonanal (0.455), Hexanal (−0.442), Octanal (0.437), Benzaldehyde (−0.421), Styrene (0.410), Heptane (−0.397)
PC4 Aldehydic degradation products	Butylated Hydroxytoluene (0.440), Benzaldehyde (0.432), Phenol (−0.425), 2,4-di-tert-butylphenol (0.418), Octanal (−0.407), Hexanal (0.395), 1-methoxypropan-2-ol (−0.383), Nonanal (0.379), Styrene (−0.366), Diethyl phthalate (0.351)
PC5 Phenolic antioxidants and stabilisers	Phenol (0.476), o-xylene (−0.386), 2,4-di-tert-butylphenol (0.374), Butylated Hydroxytoluene (−0.362), Toluene (0.359), Bis(2-ethylhexyl) phthalate (−0.345), Diethyl phthalate (0.332), Dibutyl phthalate (−0.321), Hexanal (0.309), Nonanal (0.301)

Note: Positive loadings indicate that the compound increases together with the respective principal component, while negative loadings indicate an opposite trend. The sign itself is arbitrary due to axis orientation in PCA; the magnitude of the loading is the key measure of how strongly each compound contributes to the component.

## Data Availability

The original contributions presented in this study are included in the article/Appendix A. Further inquiries can be directed to the corresponding author.

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
