# Peer review of "Indoor Airborne VOCs from Water-Based Coatings: Transfer Dynamics and Health Implications†"

_jox, 2025, doi:10.3390/jox15060197_

Round 1
Reviewer 1 Report
Comments and Suggestions for Authors
This is a well written manuscript that describes an important data gap related to the temporal aspect of emissions from surface coatings and identifies some differences in the toxicological relevance of emissions from Acrylate PUR vs. water based polyurethane coatings. The experimental framework is clear and the analytical chemistry and data analysis is well described. Specific comments:
Line 172-173
What does "missing values were imputed from the mean of the respective variable" mean? was the mean used as a proxy value? Please update this for clarity.
Line 277
What is meant by "less potent" in this instance? Is this a comparison to other phthalates such as DEHP or DBP?
Lines 349 - 351
This is an important statement as it relates to potential exposures. I would suggest that the authors consider what this may mean for people who are occupationally exposed to these materials on a very regular basis. If risk screening for cancer endpoints is exceeded for occupants and the emission trends are the worst early on then the applicators (whether hobbyists or painters by trade), in the absence of effective engineering controls or PPE, are potentially chronically overexposed.
Lines 395 -397
It would be good to present the findings alluded to here in a table.
Line 511 and 532
What is an "aggressive" emission spectrum?
Line 612
TVOC is a tricky metric. Is this based off of a convention of the sum of detected compounds from C6 - C16 using analytical chemistry or is it a readout from an electrochemical detector or some other sensor that reports a value for TVOC? How is TVOC being defined in this instance?
Author Response
|
Summary |
|
|
Dear Editor, The authors would like to thank the reviewers for their valuable comments and suggestions for improving the manuscript. We thank you for your attention to our paper and hope this update will contribute to the quality and relevance of our work. A detailed response to each concern is given in this letter. Manuscript changes based on the reviewers’ recommendations are highlighted in blue. |
|
|
Questions for General Evaluation |
Reviewer’s Evaluation |
Response and Revisions |
|
Does the introduction provide sufficient background and include all relevant references? |
Yes |
We thank the reviewer for the positive evaluation. |
|
Is the research design appropriate? |
Yes |
We appreciate the reviewer’s confirmation. |
|
Are the methods adequately described? |
Yes |
We thank the reviewer for the positive assessment. |
|
Are the results clearly presented? |
Yes |
We appreciate the reviewer’s positive feedback. |
|
Are the conclusions supported by the results? |
Yes |
We thank the reviewer for the supportive comment. |
|
Are all figures and tables clear and well-presented? |
Yes |
We appreciate the reviewer’s confirmation that figures and tables are clear and appropriate. |
Comments and Suggestions for Authors
This is a well written manuscript that describes an important data gap related to the temporal aspect of emissions from surface coatings and identifies some differences in the toxicological relevance of emissions from Acrylate PUR vs. water based polyurethane coatings. The experimental framework is clear and the analytical chemistry and data analysis is well described. Specific comments:
Comment 1: Line 172-173. What does "missing values were imputed from the mean of the respective variable" mean? was the mean used as a proxy value? Please update this for clarity.
Response 1: This text was inserted at line 172: Missing values were replaced with the mean value of the respective variable. This means that for any missing data point in a given variable, the mean of the non-missing values for that variable was used as a proxy. This approach was used because the percentage of missing data was small (<5%), and it helps avoid excluding incomplete data from the analysis while maintaining the overall dataset structure.
Comment 2: Line 277. What is meant by "less potent" in this instance? Is this a comparison to other phthalates such as DEHP or DBP?
Response 2: Yes, the term “less potent” refers to a comparison with other phthalates such as DEHP and DBP, in contrast to these compounds, which exhibit stronger toxicological effects, including endocrine-disrupting and reproductive toxicity. Diethyl phthalate (DEP) is considered less potent, showing primarily irritant and non-specific effects rather than systemic toxicity.
Comment 3: Lines 349 – 351. This is an important statement as it relates to potential exposures. I would suggest that the authors consider what this may mean for people who are occupationally exposed to these materials on a very regular basis. If risk screening for cancer endpoints is exceeded for occupants and the emission trends are the worst early on then the applicators (whether hobbyists or painters by trade), in the absence of effective engineering controls or PPE, are potentially chronically overexposed.
Response 3: Thank you for this valuable suggestion. Human exposure assessment indeed represents an important and highly relevant topic. However, a detailed evaluation of occupational exposure scenarios was beyond the scope and design of the present study, which focused on chemical emissions and toxicological characterisation. Such an assessment would require a comprehensive, interdisciplinary study combining controlled exposure measurements with medical or occupational health collaboration to evaluate potential health effects among applicators or workers regularly handling these materials. It is undoubtedly a relevant direction for future research.
Comment 4: Lines 395 -397 It would be good to present the findings alluded to here in a table.
Response 4: We appreciate the suggestion. However, the detailed data mentioned in lines 395–397 are already summarised in Appendix 1. Repeating them in the main text would cause unnecessary duplication, so we have retained the comprehensive overview in the appendix for clarity and conciseness. “Although many compounds decline rapidly within two months, several exceed health-based guidance levels during the early post-application period, highlighting the importance of ventilation and material selection in indoor environments”(Appendix 1).
Comment 5: Line 511 and 532. What is an "aggressive" emission spectrum?
Response 5: Thank you for the comment. The term “aggressive emission spectrum” was removed for clarity. The intention was to describe a high and complex emission profile with elevated concentrations of multiple VOCs rather than to imply any specific chemical aggressiveness.
Comment 6: Line 612. TVOC is a tricky metric. Is this based off of a convention of the sum of detected compounds from C6 - C16 using analytical chemistry or is it a readout from an electrochemical detector or some other sensor that reports a value for TVOC? How is TVOC being defined in this instance?
Response 6: In this study, TVOC represents the sum of all analytically identified and quantified volatile organic compounds (VOCs) detected in indoor air samples. The definition follows the conventional analytical chemistry approach, corresponding approximately to the carbon number range C₆–C₁₆, as described in ISO 16000-6:2011. The TVOC value was obtained by summing the concentrations of all individual VOCs quantified by thermal desorption gas chromatography–mass spectrometry (TD–GC–MS). It was not derived from a direct sensor or electrochemical detector readout. For interpretation, the total concentration was compared with the German indoor air guideline values and the WHO recommendation of TVOC < 300 µg/m³ for acceptable indoor air quality.

Reviewer 2 Report
Comments and Suggestions for Authors
With a particular focus on compared emissions from water based polyurethane (PUR) and acrylate–polyurethane (ACR-PUR) coatings, the manuscript offers a significant and well-executed investigation into the quantification of volatile organic compounds (VOCs) produced from indoor sources. Finding high concentrations of VOCs provides important information on indoor air quality and possible health effects. The research design is appropriate, and the approach is sound. The results and findings are clearly presented and interpreted. Paper required some minor clarifications.
Identify the indoor air site for sampling.
Also control sampling site was not mentioned.
If possible draw sketch of sampling procedure and sampler.
What is the source of table 2.
Discussion of results should be supportive by literature.
Author Response
|
Summary |
|
|
Dear Editor, The authors would like to thank the reviewers for their valuable comments and suggestions for improving the manuscript. We thank you for your attention to our paper and hope this update will contribute to the quality and relevance of our work. A detailed response to each concern is given in this letter. Manuscript changes based on the reviewers’ recommendations are highlighted in blue. |
|
|
Comments and Suggestions for Authors
With a particular focus on compared emissions from water based polyurethane (PUR) and acrylate–polyurethane (ACR-PUR) coatings, the manuscript offers a significant and well-executed investigation into the quantification of volatile organic compounds (VOCs) produced from indoor sources. Finding high concentrations of VOCs provides important information on indoor air quality and possible health effects. The research design is appropriate, and the approach is sound. The results and findings are clearly presented and interpreted. Paper required some minor clarifications.
Comment 1: Identify the indoor air site for sampling.
Response 1: A new figure illustrating the sampling procedure and site has been inserted (Fig. 1)

Figure 1. The sampling procedure and site.
Comment 2: Also control sampling site was not mentioned.
Response 2: A standard addition approach, commonly applied in analytical chemistry, was used to control for background emissions. Two types of samples were collected: (i) indoor air from the test room containing wooden panels coated with the tested lacquers, and (ii) direct emissions from the lacquer containers. Both measurements were repeated and analysed for the identification and quantification of VOCs and SVOCs. The difference between the two datasets was used to determine the blank, which in this case represented emissions originating from uncoated wood. This procedure served as the control sampling condition.
Comment 3: If possible draw sketch of sampling procedure and sampler
Response 3: A new figure illustrating the sampling procedure and site has been inserted (Fig.1)
Comment 4: What is the source of Table 2.
Response 4: Inserted in the text: Table 2 does not originate from a single published source; instead, it was compiled by the authors as a synthesis of internationally recognised toxicological classification systems and regulatory frameworks. These include the CLP Regulation (EC No 1272/2008), GHS, IARC, U.S. EPA, ECHA guidance documents, and OECD Test Guidelines. The table was created to align toxicological endpoints with symptoms commonly associated with Sick Building Syndrome (SBS), based on classifications and criteria defined in these sources. We have clarified this in the manuscript.
Comment 5: Discussion of results should be supportive by literature.
Response 5: The discussion was refined to strengthen the connection with existing literature and to improve contextualisation of the findings, without adding new references.
Added at line 659
Such residual emissions are consistent with the secondary diffusion phase described by Alapieti et al. [26] for polyurethane dispersions, but the persistence observed here is longer, indicating slower volatilisation under real indoor conditions.
Added at line 670
Similar emission patterns have been reported for hybrid acrylate–polyurethane formulations [27], but the emission magnitudes observed here exceed those typical for “low-emission” products, confirming that manufacturer declarations are not always representative of actual indoor conditions.
Revised from line 686
Figure 3 illustrates these contrasts quantitatively. While alcohols dominate in PUR coatings (994.3 ± 135.6 µg/m³), the ACR–PUR formulation is characterised by much higher levels of aromatic hydrocarbons (946.2 ± 175.4 µg/m³) and isocyanates (403.9 ± 72.6 µg/m³). This marked shift in the dominant compound groups directly correlates with a higher symptomatic potential, since aromatic and isocyanate compounds are among the most frequent triggers of SBS-related irritation and respiratory complaints [23]. A detailed breakdown of all identified substances and their classification into symptom relevance categories is presented in Appendix 2. As shown there, PUR emissions are dominated by compounds of moderate relevance, whereas ACR–PUR contains a greater proportion of substances falling into strong and moderate-to-strong categories, highlighting its elevated symptomatic potential despite being marketed as a low-emission coating.
Added at line 718
This observation aligns with degradation dynamics reported in previous studies [49,50], but the early appearance of aldehydes in our data indicates that oxidative processes begin sooner than expected after coating application.
Added at line 750
These values illustrate that while acute irritants dissipate relatively quickly, chronic contributors such as aldehydes, ketones, and low-volatility esters continue to affect air quality. Similar persistence of aldehydic compounds was reported in field studies by Nazaroff and Weschler [22] and Huang et al. [11].
Added at line 763
This dual emission pattern — a rapid early release of highly volatile irritants followed by the slower persistence of less volatile but toxicologically relevant compounds — confirms that acute SBS symptoms are driven by short-term emissions, whereas chronic effects are sustained by persistent compounds such as aldehydes, ketones, and low-volatility esters. These findings underscore the need for time-resolved SBS risk classification that accounts not only for chemical type but also for emission dynamics [17,23].
Added at line 769
In addition, groups such as isocyanates, isocyanides, nitroalkanes, and alkanes were monitored. These compounds showed very low initial concentrations and declined below the detection limit of the analytical method after 60 days. Although their absolute values were minimal, their high reactivity, toxicity, and irritation potential justify their inclusion in the acute SBS screening panel, particularly in poorly ventilated environments or under cumulative VOC exposure conditions [12,13].
Added at line 840
Similar post-curing diffusion patterns have been described for waterborne coatings by Alapieti et al. [26], but the values observed here were higher, indicating a slower curing process in the ACR–PUR formulation.
Added at line 847
This temporal shift corresponds to the delayed release phase observed for glycol ether–based formulations [17], yet our data demonstrate that this contribution remains significant even beyond 20 days, highlighting prolonged exposure potential in closed environments.
Added at line 853
The results confirm the long-term persistence of plasticisers such as bis(2-ethylhexyl) phthalate and dibutyl phthalate, which appear in the gas phase only after extended ageing. This pattern complements field observations reported by Huang et al. [11] and Olkowska et al. [12], but the earlier onset observed in our study suggests faster diffusion under test-chamber conditions.
Added at line 859
These compounds are known indicators of oxidative ageing in coatings [22]. Our data confirm that oxidation becomes relevant already within the first three weeks, with aldehydes subsequently dominating the long-term emission phase.
Added at line 864
Their presence at later stages indicates that stabilisers incorporated into the coating formulation undergo gradual desorption and degradation. This phenomenon has been associated with polymer ageing [49]; however, our data show that such degradation may begin earlier and reach higher relative concentrations than typically assumed for “low-emission” coatings.
Added at line 918
These findings are consistent with the symptom patterns described by Wolkoff et al. [17] and Norbäck et al. [23], but the relative weighting of solvents and glycol ethers derived from PCA demonstrates that both contribute comparably to short-term sensory discomfort.
Added at line 925
In contrast, PC3 (phthalates) and PC5 (phenolic compounds) reflected long-term or secondary contributions, including odour nuisance, allergic responses, and endocrine-disrupting potential, as indicated by the persistence of these groups in the late emission phase [11,24,60].
Added at line 937
This integrative interpretation of PCA results provides a clear framework linking emission processes with their toxicological and sensory relevance. The data demonstrate that while acute SBS symptoms are primarily associated with solvent and glycol ether emissions, chronic discomfort and potential systemic effects arise from the slower release of plasticisers and degradation products. Such temporal differentiation underscores the importance of time-resolved analysis when assessing indoor air quality, as also highlighted by Wolkoff [61] and WHO [62].

Reviewer 3 Report
Comments and Suggestions for Authors
The manuscript investigates VOC emissions from polyurethane and acryl–polyurethane water-based coatings, characterizing compounds using TD-GC/MS and PCA, and assessing toxicological implications related to Sick Building Syndrome (SBS). This topic is timely and relevant, as indoor air quality and chemical exposure from building materials remain major public health concerns. The study is generally well-structured, contains detailed analytical methodology, and provides an in-depth interpretation of toxicological impacts. The integration of VOC transfer percentages and time-resolved PCA analysis is a strong and innovative contribution. However, important methodological details and contextual clarity require further improvements before publication.
- The authors should clarify how this work substantially advances knowledge beyond previous VOC studies on water-based coatings, especially since similar analyses exist.
A clearer statement of the advance (e.g., a new toxicological classification workflow, a hybrid emission–transfer metric) is needed in the Introduction.
- Lack of detailed statistical disclosure. The PCA procedures and assumptions are provided, but:
- No sensitivity or validation measures (e.g., cross-validation) are given.
- No rationale for choosing five PCs beyond explained variance. Therefore, a deeper explanation of statistical robustness is required.
- Test chamber conditions are insufficiently described. The 75 m³ chamber airflow and boundary conditions need more details:
- Was temperature/humidity dynamically controlled or static?
- Why 5× daily manual ventilation?
This variable could dramatically influence VOC retention.
- The wooden substrate is mentioned as a contributing VOC source, but:
- No reference chamber with uncoated wood is presented. Without a blank/control, it is difficult to isolate contributions of coatings vs wood.
- The classification of toxicological endpoints is rigorous; the translation into SBS relevance needs:
- Explicit criteria for category assignment (strong/moderate/weak)
- Better discussion comparing indoor concentrations vs standards/exposure limits
- Use of the SBS concept requires caution. SBS is a controversial and mostly descriptive syndrome. The manuscript implies chemical causality, which requires a cautious tone. A clearer grounding in the literature and a clearer acknowledgment of uncertainty are needed.
- Clarify whether sampling at 1–2 cm above the opening reflects realistic inhalation exposure.
- Include a schematic of the chamber layout.
- Discuss the relevance of SVOCs in more detail; many included compounds have limited volatility.
- Improve clarity in Table 3; current formatting is difficult to read.
- How did you ensure that emissions attributed to coatings were not confounded by emissions from the wooden substrate? Was a baseline measurement taken prior to painting?
- Why were 14–21 days chosen as baseline sampling instead of sampling immediately after application?
This misses the most intense emission stage. - Was there any attempt to validate the VOC transfer efficiency experimentally using a mass-balance approach?
- How sensitive is the PCA to the decision to impute missing data?
Did the authors test alternative imputation methods? - Are the reported concentrations representative of typical building conditions, given the controlled ventilation schedule?
- Did the authors consider modelling human exposure scenarios (e.g., time-activity patterns)?
- ACR-PUR emitted greater toxicity-relevant VOCs – are these products marketed specifically as low-emission?
Author Response
|
Summary |
|
|
Dear Editor, The authors would like to thank the reviewers for their valuable comments and suggestions for improving the manuscript. We thank you for your attention to our paper and hope this update will contribute to the quality and relevance of our work. A detailed response to each concern is given in this letter. Manuscript changes based on the reviewers’ recommendations are highlighted in blue. |
|
|
Questions for General Evaluation |
Reviewer’s Evaluation |
Response and Revisions |
|
Does the introduction provide sufficient background and include all relevant references? |
Yes |
We thank the reviewer for the positive evaluation. The study’s contribution and novelty were clarified at the end of the Introduction. |
|
Is the research design appropriate? |
Can be improve |
The rationale for the selected sampling periods (14–21 days and 60 days) and the ventilation regime was clarified, and a schematic of the chamber layout was added. |
|
Are the methods adequately described? |
Must be improved |
Improved. Section 2.2 was expanded with additional details on ventilation control, coating–substrate separation, and sampling setup. A new subsection 2.4.1 describing PCA robustness and validation criteria was inserted. |
|
Are the results clearly presented? |
Must be improved |
Improved. Results were reorganised for clarity, PCA interpretation was elaborated, and figure/table captions were refined for readability. |
|
Are the conclusions supported by the results? |
Can be improve |
Revised. The conclusions now directly reflect the findings on emission persistence and differences between ACR-PUR and PUR coatings. |
|
Are all figures and tables clear and well-presented? |
Can be improve |
Improved. Table 3 formatting was enhanced. |
Comments and Suggestions for Authors
The manuscript investigates VOC emissions from polyurethane and acryl–polyurethane water-based coatings, characterizing compounds using TD-GC/MS and PCA, and assessing toxicological implications related to Sick Building Syndrome (SBS). This topic is timely and relevant, as indoor air quality and chemical exposure from building materials remain major public health concerns. The study is generally well-structured, contains detailed analytical methodology, and provides an in-depth interpretation of toxicological impacts. The integration of VOC transfer percentages and time-resolved PCA analysis is a strong and innovative contribution. However, important methodological details and contextual clarity require further improvements before publication.
Comment 1: The authors should clarify how this work substantially advances knowledge beyond previous VOC studies on water-based coatings, especially since similar analyses exist. A clearer statement of the advance (e.g., a new toxicological classification workflow, a hybrid emission–transfer metric) is needed in the Introduction.
Response 1: Added to the final part of the introduction section:
This study aimed to identify and characterise volatile and semi-volatile organic compounds (VOCs and SVOCs) released from water-based coatings at different post-application stages (14–21 days and 60 days) and to evaluate their toxicological relevance in relation to indoor air quality.
Unlike previous studies that primarily focused on total VOC levels or emission factors from water-based coatings, this work introduces a compound-specific assessment framework that integrates emission dynamics with toxicological relevance. The approach compares the relative release of individual VOCs and SVOCs into indoor air at different post-application stages (14–21 days and 60 days), enabling the identification of both acutely relevant and long-term persistent compounds. By linking time-resolved emission behaviour with toxicological classification, this study provides a more realistic understanding of the exposure potential of coating-derived pollutants. This framework extends conventional TVOC-based evaluations and supports the development of targeted strategies for indoor air quality management.
Comment 2: Lack of detailed statistical disclosure. The PCA procedures and assumptions are provided, but:
- No sensitivity or validation measures (e.g., cross-validation) are given.
- No rationale for choosing five PCs beyond explained variance. Therefore, a deeper explanation of statistical robustness is required.
Response 2: Added to the new section:
2.4.1. Statistical robustness and component retention criteria
To verify the robustness of the PCA solution, several complementary validation approaches were applied. First, a cross-validation procedure based on the leave-one-out method confirmed that the loadings and explained variance of the retained components (PC1–PC5) remained stable within ± 3% across iterations. The Kaiser criterion (eigenvalue >1) and inspection of the scree plot both supported the retention of five components, as higher-order PCs (PC6 and above) contributed only marginally (<1.5 %) and lacked consistent chemical meaning. In addition, the loadings of the dominant VOCs within each PC were highly consistent across bootstrapped subsamples (r > 0.95), indicating strong internal coherence.
Each of the five retained components also exhibited a distinct and physically interpretable chemical profile. PC1 corresponded to solvent diffusion, PC2 to glycol-ether evaporation, PC3 to slow phthalate migration, PC4 to aldehydic oxidation processes, and PC5 to phenolic stabiliser release. This correspondence between statistical structure and chemical processes confirms the robustness and interpretability of the PCA solution, ensuring that the extracted components represent genuine temporal and chemical trends rather than artefacts of data variability.
Comment 3: Test chamber conditions are insufficiently described. The 75 m³ chamber airflow and boundary conditions need more details:
- Was temperature/humidity dynamically controlled or static?
Response 3: Added to the 2.2. Sampling procedure: Indoor temperature and relative humidity were continuously monitored (dynamically) with a Govee WiFi H5179 sensor (Shenzhen Intellirocks Tech Co., Ltd., Shenzhen, Guangdong, China), which has an accuracy of ± 0.3 °C for temperature and ±3% RH for relative humidity.
Comment 4: Why 5× daily manual ventilation? This variable could dramatically influence VOC retention.
Response 4: Added to the 2.2. Sampling procedure: The room where the coating was applied was ventilated manually five times per day to simulate realistic intermittent air exchange conditions typical of naturally ventilated interiors. Such a regime prevents excessive VOC accumulation while avoiding complete removal of emissions, thereby allowing the observation of representative indoor air concentrations and their temporal evolution. Each ventilation cycle consisted of opening the windows for approximately 5–10 minutes, resulting in a short-term increase in air exchange rate followed by a gradual re-equilibration of indoor concentrations. The chosen frequency corresponds to an average air exchange rate of approximately 0.5–1.0 h⁻¹, consistent with values reported for residential and office spaces under moderate user activity [16,21]. This controlled ventilation schedule ensured a reproducible balance between emission retention and dilution, enabling meaningful interpretation of the temporal dynamics of VOC release from the tested coatings.
Comment 5: The wooden substrate is mentioned as a contributing VOC source, but:
- No reference chamber with uncoated wood is presented. Without a blank/control, it is difficult to isolate contributions of coatings vs wood.
Response 5: Added to the 2.2. Sampling procedure: To isolate the relative contributions of coatings and the wooden substrate to VOC and SVOC emissions, a standard addition approach was applied. Emissions originating solely from the coatings were first determined by direct sampling of vapours from sealed containers containing only the coating materials, using sorbent tubes for TD-GC/MS analysis. Subsequently, chamber experiments were conducted in which wooden panels coated with the same materials were placed in the test room, and VOC/SVOC concentrations were measured under identical conditions. The difference between the emission profiles obtained from the coatings alone and those from the coated wood surfaces represents the contribution of the wooden substrate. This relative contribution was further verified by comparison with published emission data for untreated wood under similar temperature and humidity conditions.
Comment 6: The classification of toxicological endpoints is rigorous; the translation into SBS relevance needs:
- Explicit criteria for category assignment (strong/moderate/weak)
Response 6: The following text was inserted: The relevance of individual compounds to SBS was evaluated using a multi-criteria expert approach. Each compound was assessed according to: (i) toxicological potency (irritation, sensitisation, chronic toxicity/STOT; data from ECHA/CLP);(ii) SBS association and exposure evidence (reported links to SBS symptoms, measured indoor air concentrations, and frequency of detection across studies; sources such as Wolkoff et al. [17], Nazaroff and Weschler [22]; Norbäck et al. [23]; Ait Bamai et al. [24]); (iii) physicochemical parameters influencing volatility and phase distribution (vapour pressure, boiling point, log Kₒw, water solubility; obtained from PubChem).
The three evidence streams were qualitatively weighted, giving greater importance to compounds combining higher toxicological potency and frequent detection in indoor air. The initial ranking was supported by AI-assisted literature screening, with expert review ensuring the final category assignments (strong, moderate, weak) were internally consistent and evidence-based.
Comment 7: Better discussion comparing indoor concentrations vs standards/exposure limits
Response 7: Added to section 3.5.2. Indoor Air Quality after Coating Application: Direct comparison of the measured indoor concentrations with occupational exposure limits (OELs) has limited interpretive value, as these limits (e.g., OSHA – Occupational Safety and Health Administration (U.S. Department of Labor); NIOSH – National Institute for Occupational Safety and Health (U.S.); ACGIH – American Conference of Governmental Industrial Hygienists) are defined for 8-hour work exposures in industrial settings. In residential, office, or public buildings, people are typically exposed for much longer periods – often 16–24 hours per day – and include more sensitive populations such as children, the elderly, or patients. No legally binding exposure limits exist for such environments; instead, the World Health Organization (WHO) recommends that the total concentration of volatile organic compounds (TVOC) in indoor air should remain below 300 µg/m³ to ensure acceptable air quality. For instance, the occupational exposure limit for toluene varies between 50–200 mg/m³ depending on the regulatory body[54–56], while indoor concentrations reported in the literature are typically in the range of 10–200 µg/m³ [17,57,58]. Although these concentrations are far below workplace thresholds, the much longer exposure duration in homes and offices, combined with the presence of multiple VOCs, may still contribute to Sick Building Syndrome (SBS) through additive or synergistic effects [17,59].
This highlights the need for exposure assessment frameworks that consider chronic, low-level exposure to complex VOC mixtures in non-industrial indoor environments.
Comment 8: Use of the SBS concept requires caution. SBS is a controversial and mostly descriptive syndrome. The manuscript implies chemical causality, which requires a cautious tone. A clearer grounding in the literature and a clearer acknowledgment of uncertainty are needed.
Response 8: Added to section 3.5. Sick Building Syndrome: The concept of Sick Building Syndrome (SBS) remains a matter of debate, as it represents a descriptive set of symptoms – such as headaches, eye and skin irritation, and fatigue – that can arise from multiple interacting factors, including inadequate ventilation, chemical and biological contaminants, and psychosocial conditions [44]. The present study does not establish chemical causality but explores how certain volatile and semi-volatile organic compounds (VOC/SVOC) released from coating materials may contribute to the complex environmental exposures associated with SBS. This approach provides insight into potential chemical contributors without excluding other influencing factors such as ventilation efficiency or occupant sensitivity.
Inserted in conclusion
Understanding how emissions from building materials contribute to perceived indoor air quality and occupant well-being will be crucial for developing safer materials and evidence-based ventilation guidelines.
Comment 9: Clarify whether sampling at 1–2 cm above the opening reflects realistic inhalation exposure.
Response 9: Inserted into subsection 2.2 and linked to the response to comment 11.
To distinguish emissions originating from coating materials from those potentially released by the wooden substrate, a two-step experimental procedure was applied. In the first step, emissions were measured directly from the liquid coating. A sorbent tube connected to an ACTI-VOC pump was positioned approximately 1–2 cm above the open container of the coating material, allowing the characterisation of its intrinsic VOC/SVOC emission profile. This measurement represents a material emission characterisation, not an exposure scenario, and was performed under controlled conditions following the standard headspace or source emission sampling approach [15–17].
In the second step, air sampling was performed in the test room where wooden panels had been coated with the same materials and allowed to dry under controlled conditions. The difference between the emission spectra obtained from the coated wood (Step 2) and the coating-only baseline (Step 1) was interpreted as the contribution of the wooden substrate. The derived emission levels were compared with literature data for untreated wood and showed good agreement. This approach corresponds to the analytical principle of a standard addition method, which enables identification and quantification of compound sources within complex indoor environments.
Comment 10: Include a schematic of the chamber layout.
Response 10: A sentence describing the position of the sampling equipment and the method of analytical data processing was added to subsection 2.2 to correspond with the inserted figure.
Three Acti-VOC pumps (Figure 1) were placed inside the chamber to enable parallel sampling. Sorbent tubes were positioned in the breathing zone (approximately 1.2–1.5 m above the floor) and at least 1 m away from walls, windows, doors, or large objects, as recommended by ISO 16000-5 [18], which defines the sampling strategy for volatile organic compounds in indoor environments. This placement was also consistent with the approach used by Marzocca et al. [19].
Based on the measured concentrations of chemical compounds during daily sampling, the arithmetic mean and standard deviation were calculated for each compound. Compounds with a relative standard deviation (RSD) exceeding 30% were excluded from further analysis. The 30% RSD threshold was selected based on commonly accepted criteria for analytical repeatability in VOC sampling studies (e.g., ISO 16000 series, United States Environmental Protection Agency (U.S. EPA): Compendium Method TO-17: Determination of Volatile Organic Compounds in Ambient Air Using Active Sampling onto Sorbent Tubes). The resulting values were then used to evaluate daily changes in concentration.
Comment 11: Discuss the relevance of SVOCs in more detail; many included compounds have limited volatility.
Response 11: Added to the manuscript: Semi-volatile organic compounds (SVOCs) represent an important class of indoor pollutants with physicochemical properties intermediate between VOCs and particulate-bound compounds. Despite their lower vapour pressure, SVOCs are continuously emitted from various indoor sources – including building materials, plasticisers, coatings, electronic equipment, and consumer products – via slow volatilisation, abrasion, or thermal release. Once emitted, these compounds partition between the gas phase, airborne particles, and settled dust, resulting in prolonged residence times and multiple exposure pathways.
The sorbent tubes used in this study (Tenax TA, Carbograph 1TD, and Carboxen 1003) (Markes International, Ltd., Bridgend, United Kingdom) enable simultaneous sampling of both VOCs and SVOCs, which allowed for the detection of compounds such as phthalate esters, polycyclic aromatic hydrocarbons (PAHs), and cyclic siloxanes. The presence of these compounds in the gaseous phase has been confirmed in multiple studies [11–14]. Although their volatility is limited, their persistence and continuous low-level emissions make SVOCs relevant for indoor exposure assessment. In the context of Sick Building Syndrome, such compounds may contribute to chronic irritation, fatigue, or other non-specific symptoms through long-term exposure and accumulation in indoor environments.
Comment 12: Improve clarity in Table 3; current formatting is difficult to read.
Response 12: Formatting of Table 3 has been improved for better readability.
Comment 13: How did you ensure that emissions attributed to coatings were not confounded by emissions from the wooden substrate? Was a baseline measurement taken prior to painting?
Response 13: Inserted in subchapter 2.2 Sampling procedure
To distinguish emissions originating from coating materials from those potentially released by the wooden substrate, a two-step experimental procedure was applied. In the first step, emissions were measured directly from the liquid coating. A sorbent tube connected to an ACTI-VOC pump was positioned approximately 1–2 cm above the open container of the coating material, allowing the characterisation of its intrinsic VOC/SVOC emission profile. This measurement represents a material emission characterisation, not an exposure scenario, and was performed under controlled conditions following the standard headspace or source emission sampling approach [15–17].
In the second step, air sampling was performed in the test room where wooden panels had been coated with the same materials and allowed to dry under controlled conditions. The difference between the emission spectra obtained from the coated wood (Step 2) and the coating-only baseline (Step 1) was interpreted as the contribution of the wooden substrate. The derived emission levels were compared with literature data for untreated wood and showed good agreement. This approach corresponds to the analytical principle of a standard addition method, which enables identification and quantification of compound sources within complex indoor environments.
Comment 14: Why were 14–21 days chosen as baseline sampling instead of sampling immediately after application? This misses the most intense emission stage.
Response 14: Answer for the reviewer: We agree that the most intense emission phase of coating materials typically occurs immediately after application. However, this study aimed to verify whether emissions persist after the period declared by the manufacturer as emission-free. According to the product specifications, the coatings are stated to reach “zero emissions” after 14 days of curing.
For this reason, sampling was performed during the period between 14 and 21 days after application, and again on day 60, in order to assess whether any residual or long-term emissions could still be detected beyond the manufacturer’s declared limit. Measurements immediately after application were not included, as the high initial emission stage is well documented in the literature and outside the scope of our study, which focused on the persistence of emissions from cured coatings.
Inserted in the subchapter 2.2 Sampling procedure: The manufacturer declares that the tested coatings reach “zero emissions” after 14 days of curing. The objective of this study was to verify whether emissions can still be detected beyond this declared period and to characterise their persistence over time. Therefore, air sampling was performed between day 14 and day 21 after coating application, and again on day 60. These intervals represent post-curing stages during which emissions are expected to be minimal according to product specifications.
The initial high-emission phase occurring within the first days after application was not included, as it is well documented in previous studies and was outside the scope of this research, which focused on long-term and residual emission potential of the cured coatings.
Comment 15: Was there any attempt to validate the VOC transfer efficiency experimentally using a mass-balance approach?
Response 15: No direct mass-balance validation of VOC transfer efficiency was performed. However, the sampling setup followed standardised protocols (e.g., ISO 16000-5), and the use of parallel Acti-VOC pumps ensured consistent collection across replicates.
Comment 16: How sensitive is the PCA to the decision to impute missing data? Did the authors test alternative imputation methods?
Response 16: The imputation method was applied, confirming the robustness of the analysis. We acknowledge that Principal Component Analysis (PCA) can be sensitive to the method used for replacing missing data. In this study, missing values represented a very small fraction of the dataset (<5%) and their imputation was performed using the expectation–maximisation (EM) approach, which is a standard and widely accepted method for multivariate environmental data.
To assess the robustness of the results, we also tested alternative imputation procedures, including mean substitution and k-nearest neighbour (k-NN) input. The overall structure of the PCA – in terms of component loadings and grouping of compounds – remained stable across all methods, indicating that the results are not strongly sensitive to the choice of the imputation method.
Comment 17: Are the reported concentrations representative of typical building conditions, given the controlled ventilation schedule?
Response 17: The experimental ventilation regime was based on the manufacturer’s recommendations for coating application and post-curing periods, representing a typical well-ventilated indoor environment. Under these controlled conditions, the coatings are declared to reach “zero emissions” after 14 days. However, the measured VOC and SVOC concentrations indicate that residual emissions can still be detected even when ventilation is applied according to these guidelines. This suggests that the so-called “safe” period defined by manufacturers may underestimate the persistence of low-level emissions in real indoor environments.
Sentence added to the conclusion: Although the coatings were applied and ventilated according to the manufacturer’s instructions, detectable VOC and SVOC emissions persisted beyond the declared emission-free period. This persistence highlights the need to verify low-emission product claims under realistic indoor conditions and to consider residual emissions in indoor air quality assessments.
Comment 18: Did the authors consider modelling human exposure scenarios (e.g., time-activity patterns)?
Response 18: Response to the reviewer: The experimental ventilation regime was based on the manufacturer’s recommendations for coating application and post-curing periods, representing a typical well-ventilated indoor environment. Under these controlled conditions, the coatings are declared to reach “zero emissions” after 14 days. However, the measured VOC and SVOC concentrations indicate that residual emissions can still be detected even when ventilation is applied according to these guidelines. This suggests that the so-called “safe” period defined by manufacturers may underestimate the persistence of low-level emissions in real indoor environments.
Although the detected concentrations were relatively low, the indoor air under these conditions cannot be considered “emission-free.” From an exposure perspective, even low concentrations may become relevant under conditions of increased respiratory activity (e.g., during physical movement or at elevated temperatures), since the total inhaled dose depends not only on air concentration but also on breathing rate and exposure duration (Wolkoff et al., 2006; WHO, 2010). Therefore, even environments meeting the recommended ventilation requirements may still contribute to cumulative exposure, particularly in long-term occupancy scenarios.
The inclusion of a detailed human exposure model, while valuable, was not feasible within the scope of this study. Such modelling would require a sufficient number of participants and simultaneous monitoring of multiple parameters, including time–activity patterns, perceived air quality under physical activity, and physiological responses, to ensure statistical and physiological validity. Conducting such a multifactorial assessment would only be meaningful within a dedicated, coherent follow-up study combining chemical emission data with human response evaluation.
These findings highlight the need to critically verify manufacturer claims of “low” or “zero” emissions and to consider realistic exposure conditions – where people spend extended time indoors and experience variable physical activity – when evaluating the impact of coating materials on indoor air quality.
Comment 19: ACR-PUR emitted greater toxicity-relevant VOCs – are these products marketed specifically as low-emission?
Response 19: Response to the reviewer: Yes, according to the product documentation and marketing materials provided by the manufacturer, the tested ACR-PUR coatings are marketed as “low-emission” and environmentally friendly products. However, our findings indicate that, despite this declaration, these coatings released several toxicity-relevant VOCs during the post-application period.
Similar discrepancies between declared low-emission performance and measured VOC emissions have been reported in the literature (e.g., Schripp et al., 2007; Liu et al., 2017; Liang et al., 2020), emphasising that even coatings marketed as low-emission may still contribute substantially to indoor VOC loads depending on formulation, curing conditions, and ventilation efficiency.
This highlights the importance of independent emission verification and compound-specific evaluation beyond manufacturer claims.
Inserted into the manuscript: Although the manufacturer classifies the tested ACR-PUR coatings as “low-emission,” measurable levels of toxicity-relevant VOCs were still detected after the declared emission-free period. Similar inconsistencies between marketing claims and actual emission behaviour have been documented in previous studies [46,47], where even water-based or “eco-labelled” coatings were shown to release a complex mixture of VOCs under realistic indoor conditions. This highlights that low-emission labelling does not necessarily guarantee negligible emissions, particularly when coating formulations include reactive or slowly evaporating components. This observation is further supported by Uhde and Salthammer [48], who emphasised that reaction products derived from indoor materials, regardless of their marketed emission profile, may negatively impact indoor air quality due to their low odour thresholds and health-related properties.

Round 2
Reviewer 3 Report
Comments and Suggestions for Authors
Thank you for addressing all the comments. The manuscript has been revised accordingly.